# MULTI-AGENT DEBATE WITH MEMORY MASKING

**Hongduan Tian[1]  Xiao Feng[1]  Ziyuan Zhao[2]  Xiangyu Zhu[2]  Rolan Yan[2]  Bo Han[1] [†]**
[1]TMLR Group, Hong Kong Baptist University  [2]Search Algorithm Group, WeChat, Tencent
`{cshdtian, xiaofeng, bhanml}@comp.hkbu.edu.hk`
`{joshuazhao, xiangyuzhu, rolanyan}@tencent.com`

## ABSTRACT

Large language models (LLMs) have recently demonstrated impressive capabilities in reasoning tasks. Currently, mainstream LLM reasoning frameworks predominantly focus on scaling up inference-time sampling to enhance performance. In particular, among all LLM reasoning frameworks, *multi-agent debate* (MAD), which employs multiple LLMs as agents to perform reasoning in the way of multi-round debate, has emerged as a powerful reasoning paradigm since it allows agents to access previous memories to alleviate fallacious content and refine their reasoning iteratively in each debate round. However, although MAD significantly improves the reasoning capabilities of LLMs, in this paper, we observe that there remain erroneous memories, and LLM agents are vulnerable to these erroneous memories. To explore this phenomenon, we provide a theoretical insight that the performance of MAD is highly dependent on the quality of memories derived from the previous debate, indicating that the existence of erroneous memories poses a threat to the performance of MAD. To address this problem, we introduce a simple yet effective multi-agent debate framework, *multi-agent debate with memory masking* (MAD-$M^2$), to improve the robustness of MAD by allowing LLM agents to mask erroneous memories from the previous debate round at the beginning of each debate round. In this way, MAD-$M^2$ can polish the contextual information before each debate round by preserving informative and meaningful memories while discarding the erroneous memories. Extensive experiments on mainstream math reasoning and language understanding benchmarks demonstrate that MAD-$M^2$ can identify the erroneous memories and achieve better performance in reasoning than MAD. The code of MAD-$M^2$ is available at: https://github.com/tmlr-group/MAD-MM.

## 1 INTRODUCTION

Large language models (LLMs) have shown impressive reasoning capabilities in a range of tasks and have attracted increasing attention in this field. Typically, in LLMs, provided with elaborately designed instructions (Ouyang et al., 2022), LLMs can complete the tasks with exceptional instruction-following ability. Moreover, it has also been demonstrated that the reasoning capability can be further enhanced through fine-grained reasoning tricks (e.g., stepwise reasoning (Wei et al., 2022b; Kojima et al., 2022; Zhou et al., 2023) and reflection (Madaan et al., 2023)) with abundant demonstrations (Brown et al., 2020). Despite such an impressive capability, a challenge in LLM reasoning is invalid facts and fallacious content. Although some previous works propose to alleviate the problem via multiple sampling (Wang et al., 2023; Taubenfeld et al., 2025), due to the insufficient generation diversity issue (Si et al., 2025; Hayati et al., 2023), LLMs may be stuck in their own biases, thereby limiting the efficacy of the test-time scaling strategies (Snell et al., 2024; Brown et al., 2024).

Inspired by *The Society of Mind* (Minsky, 1986), *Multi-agent debate* (MAD (Du et al., 2023)), which employs multiple LLMs as agents to perform reasoning via a multi-round debate, has recently emerged as an effective approach to overcome the aforementioned two limitations. Specifically, in each debate round (except for the first round), agents are required to generate answers based on their critical evaluation of all memories from the previous round. In this case, MAD improves the reasoning capability of LLMs mainly from two perspectives. On the one hand, in MAD, agents have access to all previous memories and generate new answers based on their critical evaluation of memories. Thus, the invalid and fallacious content can be identified and corrected. On the other

---

[†]Correspondence to Bo Han (bhanml@comp.hkbu.edu.hk).

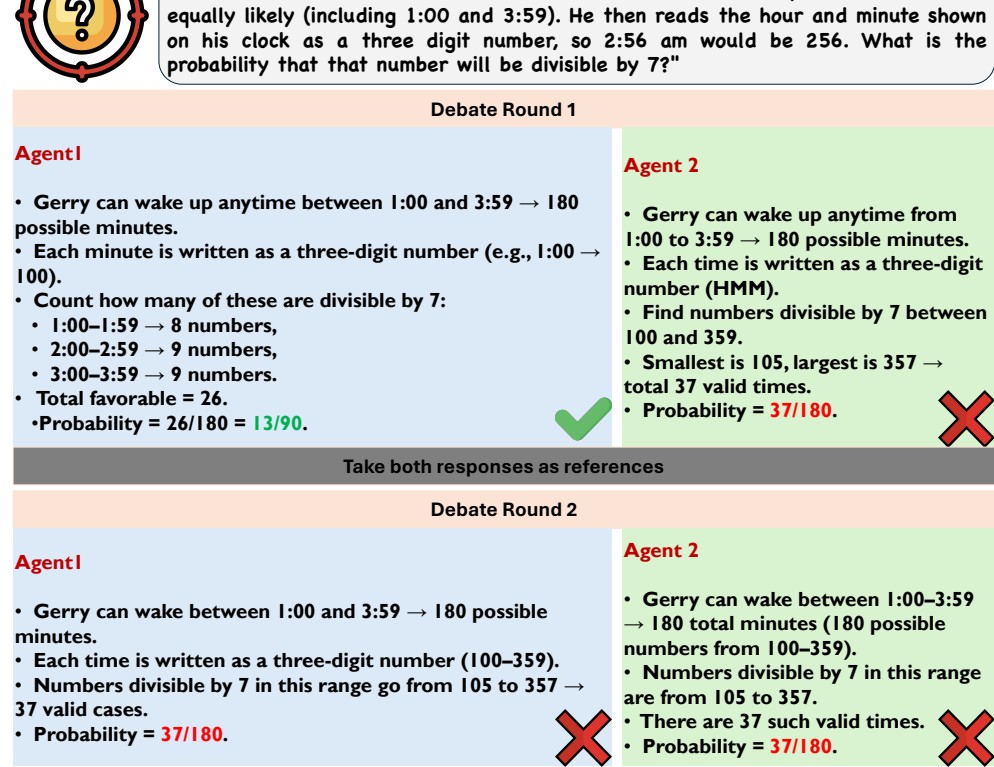

Figure 1: **An illustration of the effects of erroneous memories.** The example is a real case picked from the MATH dataset. In MAD, all memories in the previous debate round are considered in the next debate round. However, the memories from the previous round may include erroneous reasoning responses (cf. the responses of debate round 1 in the figure). The erroneous memory may misguide the agent, which was correct, and result in the wrong final answer (cf. Agent 1 in the debate round 2).

hand, by employing previous memories as in-context information for the next debate round, agents may get rid of their inherent bias (Panickssery et al., 2024) and obtain better answers regarding the query. Nevertheless, a concern about MAD remains: *Are LLM agents robust to erroneous memories?* For example, as shown in Fig. 1, memories derived from the previous debate round may be incorrect (e.g., Agent 2 in the debate round 1), and the agent, which takes all memories into consideration, will be misguided and generate the incorrect answers (e.g., Agent 1 in the debate round 2), thereby undermining the reasoning performance. As far as we know, this concern remains underexplored.

To achieve a comprehensive understanding of MAD, we provide an analysis from the perspective of the probability that MAD can correctly infer the answer to a given query. The results indicate that the multi-agent debate framework, under mild assumptions, is vulnerable to erroneous memories. Specifically, the performance of LLM agents in a debate round closely depends on the number of wrong memories in the previous debate round. When the number of wrong memories from the previous debate round increases, the reasoning capability of LLMs degrades correspondingly. According to our theoretical analyses, in both easy and hard reasoning scenarios, enhancing the robustness of LLM agents to erroneous memories consistently improves the performance of MAD.

Motivated by the empirical observation and theoretical analysis, we propose a simple yet effective method, *multi-agent debate with memory masking* (MAD-M$^2$), to enhance the robustness of MAD to erroneous memories by allowing LLMs to mask the incorrect memories in the previous debate round depending on the evaluations on those memories. Specifically, at each debate round (except for the first debate round), the memories derived from the previous debate round will be critically evaluated. Then, a binary mask vector will be generated according to the evaluations to filter the memories. Then, LLM agents are required to generate new responses based on the preserved memories. The final answer will be selected from a set of responses by performing majority voting in the final round. Intuitively, the reasoning in each round resembles an in-context learning process (Wei et al., 2022a).

In this case, erroneous memories are equivalent to poor-quality demonstrations that may distract LLMs (Shi et al., 2023), thereby leading to poor performance. Thus, MAD-M$^2$ improves MAD's capability by filtering out fallacious content to improve contextual information (Mei et al., 2025).

To validate the effectiveness of MAD-M$^2$, mainstream mathematical reasoning (e.g., GSM8K (Cobbe et al., 2021), MATH (Hendrycks et al., 2021), and AIME24&25) and language understanding (e.g., MMLU-Pro (Wang et al., 2024)) benchmarks are adopted for evaluation. Empirically, MAD-M$^2$ generally achieves better performance than naive MAD (Du et al., 2023) on various reasoning tasks.

Overall, our contribution in this paper can be primarily summarized as the following four aspects:

- We find that LLM agents in the conventional MAD framework are vulnerable to erroneous memories derived from the last debate round and may be misled to incorrect reasoning responses (c.f. Fig. 1).
- In Section 2, we demonstrate the vulnerability of the MAD framework from a mathematical perspective. Theoretically, compared to simply increasing the number of sampling/agents, filtering out erroneous memories in the last debate consistently improves the performance of MAD in all cases. As far as we know, ours is the first work to discuss this phenomenon in multi-agent debate.
- In Section 3, to alleviate the detriment above, we propose a simple yet effective multi-agent debate framework, *multi-agent debate with memory masking* (MAD-M$^2$), to enhance the capability of MAD by critically evaluating memories in the last debate and masking those incorrect memories.
- In Section 4, we empirically evaluate MAD-M$^2$ on both mathematical and language understanding benchmarks and demonstrate that MAD-M$^2$ generally achieves better performance than MAD.

## 2 AN OVERVIEW OF MULTI-AGENT DEBATE FRAMEWORK

In this section, we first introduce an overview and formulate the workflow of the conventional multi-agent debate framework (Du et al., 2023). Then, we theoretically analyze the probability of MAD successfully inferring the answer to a given query. The results further demonstrate that the naive MAD framework is vulnerable to erroneous memories derived from the last round of debate.

### 2.1 PROBLEM FORMULATION OF MULTI-AGENT DEBATE

Multi-agent debate (MAD) (Du et al., 2023) is a powerful reasoning paradigm that infers the answers to questions through multiple rounds of interactions among multiple LLM agents. Specifically, in the typical multi-agent debate framework, LLMs are first instructed to perform reasoning on the queries independently at the initial round. Then, from the second round, agents are required to critically evaluate the memories of all agents in the last debate round to generate new responses. After a multi-round debate, the final answer is obtained by performing majority voting on a set of candidates.

Consider a typical multi-agent debate framework (Du et al., 2023) composed of a total of $N_{\text{round}}$ debate rounds and a set of $N_{\text{a}}$ LLM agents $\mathcal{A} = \{A_{\theta_1}, A_{\theta_2}, ..., A_{\theta_{N_{\text{a}}}}\}$ that are parameterized with parameters $\theta_i$, respectively. Consider a query prompt $x^{\text{test}} \in \mathcal{X}$ that includes both task instructions and the question, where $\mathcal{X} = \{x_1^{\text{test}}, x_2^{\text{test}}, ..., x_{N_t}^{\text{test}}\}$ is a discrete set of $N_t$ query prompts. Let $\mathcal{M}_r = [A_{\theta_1}(x^{\text{test}}, \mathcal{M}_{r-1}), A_{\theta_2}(x^{\text{test}}, \mathcal{M}_{r-1}), ..., A_{\theta_{N_{\text{a}}}}(x^{\text{test}}, \mathcal{M}_{r-1})]$, where $1 \leq r \leq N_{\text{round}}, \mathcal{M}_0 = \emptyset$, denote the memory set derived from the $r$-th debate round. Then, for each LLM agent $A_{\theta_i} \in \mathcal{A}$, given the memories $\mathcal{M}_r$ and prompt $x^{\text{test}}$, the response generation in the next round is formulated as:

$$A_{\theta_i}(x^{\text{test}}, \mathcal{M}_r) = \hat{x}_{1:L}, \text{where } \hat{x}_l = \arg\max_x P(x|\hat{x}_{<l}; x^{\text{test}}; \mathcal{M}_r; \theta_i), \qquad (1)$$

where $\hat{x}_l$ denotes the $l$-th word prediction of the response, $\hat{x}_{<l}$ denotes the previous word predictions of $\hat{x}_l$, and $L$ denotes the length of the response. The response $\hat{x}_{1:L}$ is a sequence with the length of $L$.

### 2.2 MAD IS VULNERABLE TO ERRONEOUS MEMORIES

In the typical multi-agent debate framework (cf. Eq. (1)), LLM agents are required to generate new responses based on the memories derived from the previous debate round. Thus, intuitively, erroneous memories in the previous debate round may potentially undermine the performance of MAD (cf. Fig. 1). However, as far as we know, few works have discussed this issue in detail. Thus, in this section, we provide a discussion to explore the effect of erroneous memories on the MAD framework.

#### 2.2.1 A THEORETICAL PERSPECTIVE FOR UNDERSTANDING OF MAD

In this section, we first provide a theoretical understanding of MAD. As a comparison, we here adopt CoT-SC (Wang et al., 2023) as the counterpart of MAD. From the perspective of the reasoning

paradigm, both CoT-SC and MAD can be seen as the test-time scaling reasoning framework (Snell et al., 2024; Brown et al., 2024), which improves the robustness and the performance of reasoning by sampling multiple responses for a single query. The main difference between the two reasoning paradigms is that CoT-SC performs reasoning once and obtains the answer by voting for the majority among a set of responses, while MAD is formulated as the multi-round reasoning among multiple agents, where the final answer is voted from responses generated by agents in the final debate round.

**Assumption 2.1.** *Assume that the probability that an agent independently generates the correct answer only with the given query is $p \in [0, 1]$, while the probability that the agent generates correct answers to the query based on the previous memories is assumed to be $e^{-\alpha N_e}$, where $N_e$ denotes the number of erroneous memories and $\alpha \in \mathbb{R}^+$ is a constant coefficient that indicates the robustness of agents to erroneous memories. The reasoning is deemed as successful when the number of correct answers $N_{\text{cor}}$ in the final round satisfies $N_{\text{cor}} > \frac{N_{\text{res}}}{2}$, where $N_{\text{res}}$ denotes the number of responses.*

**Remark 1.** *In Assumption 2.1, we formulate the probability that a single LLM agent generates the correct answer to the given query based on the previous memories as $e^{-\alpha N_e}$, where $N_e \in \{0, 1, 2, ..., N_a\}$ denotes the number of erroneous memories. Two aspects are considered here. On the one hand, in an ideal case where the number of erroneous memories is 0, the probability of the agent generating the correct reasoning in the next debate round will be 1. On the other hand, in the case where all memories are erroneous (i.e., $N_e = N_a$), the probability of the agent generating a correct response to the query will degrade to $e^{-\alpha N_a} > 0$. This aligns with the situation that agents can retain a non-zero probability of generating correct reasoning even if all memories are erroneous.*

**Proposition 2.2** (**CoT-SC**). *Consider a total number of $N_{\text{sc}}$ independent responses generated in the way of CoT-SC. With Assumption 2.1, the probability that the final answer is correct is bounded by:*

$$P(N_{\text{cor}} > \frac{N_{\text{sc}}}{2}) \leq \exp\left(-2N_{\text{sc}}(\frac{1}{2} - p)^2\right), \qquad p < \frac{1}{2},$$

$$P(N_{\text{cor}} > \frac{N_{\text{sc}}}{2}) \geq 1 - \exp\left(-2N_{\text{sc}}(\frac{1}{2} - p)^2\right), \quad p \geq \frac{1}{2}.$$

*The corresponding lower bound and upper bound of cases $p < \frac{1}{2}$ and $p \geq \frac{1}{2}$ are 0 and 1, respectively.*

**Proposition 2.3** (**MAD**). *Consider a 2-round MAD reasoning, where $N_a$ agents are involved in each debate round. With Assumption 2.1, the probability that the final answer is correct is bounded by:*

$$P(N_{\text{cor}}^{(2)} > \frac{N_a}{2}) \leq \sum_{j=0}^{j^*} \omega_j \exp\left(-2N_a\left(\frac{1}{2} - e^{\alpha(j-N_a)}\right)^2\right) + \sum_{j=j^*+1}^{N_a} \omega_j, \quad e^{\alpha(j-N_a)} < \frac{1}{2},$$

$$P(N_{\text{cor}}^{(2)} > \frac{N_a}{2}) \geq \sum_{j=j^*+1}^{N_a} \omega_j\left(1 - \exp\left(-2N_a\left(\frac{1}{2} - e^{\alpha(j-N_a)}\right)^2\right)\right), \quad e^{\alpha(j-N_a)} \geq \frac{1}{2},$$

*where $\omega_j = \binom{N_a}{j} p^j (1-p)^{N_a-j}$, $j^* = \lfloor N_a - \frac{\ln 2}{\alpha} \rfloor$. For simplicity, the corresponding lower and upper bounds of situations $e^{\alpha(j-N_a)} < \frac{1}{2}$ and $e^{\alpha(j-N_a)} \geq \frac{1}{2}$ are trivial bounds 0 and 1, respectively.*

**Remark 2.** *The proofs of Propositions 2.2 and 2.3 are available in Appendix B. In Propositions 2.2 and 2.3, we investigate the capability boundary of CoT-SC and MAD in reasoning. In the case of CoT-SC, we can observe that the performance is mainly determined by the number of sampled responses (i.e., $N_{\text{sc}}$) and the capability of LLM agents in reasoning (i.e., the probability of LLM agents correctly answering the query $p$). However, in the case of MAD, the performance is explicitly determined by the number of agents (i.e., $N_a$) and the probability of LLM agents generating correct answers based on the memories in the previous round (i.e., $e^{-\alpha(N_a-j)}$). Since LLMs are vulnerable to erroneous memories, except for the initial debate round, the probability $e^{-\alpha(N_a-j)}$ fluctuates according to the number of erroneous memories in the previous round. Meanwhile, the performance is also influenced by the probability distribution of the memories in the previous debate round. In the 2-round MAD case, the distribution is formulated as a binomial distribution: $P(X = j) = \binom{N_a}{j} p^j (1-p)^{N_a-j}$.*

According to Propositions 2.2 and 2.3, both results indicate that CoT-SC and MAD rely heavily on the capabilities of agents (i.e., $p$ or $e^{-\alpha(j-N_a)}$). Specifically, the performance is determined by whether LLM agents are powerful enough to correctly perform reasoning on given tasks. Different from CoT-SC, where the capability is inherently determined, the capability of MAD is conditioned on the quality of memories from the last debate round, thereby being particularly vulnerable to erroneous

memories. Consider a case that $N_\mathrm{a} = N_\mathrm{sc}$. In the case that LLM agents can hardly generate correct reasoning trajectories (i.e., $e^{-\alpha(j-N_\mathrm{a})} < \frac{1}{2}$), the bound of MAD will become tighter (i.e., lower probability of correctly performing reasoning) compared to CoT-SC. Moreover, even for the case that agents can probably generate correct reasoning responses (i.e., $e^{-\alpha(j-N_\mathrm{a})} \geq \frac{1}{2}$), as $\omega_j$ is smaller than 1, with the mild assumption that $e^{-\alpha(j-N_\mathrm{a})}$ does not deviate much from $p$ (since MAD can easily infer the answers in this case), the lower bound of MAD is also tighter than that of CoT-SC. All discussions above imply that CoT-SC is an ideal case of MAD. These observations, to some extent, further explain why MAD generally fails to outperform CoT-SC in reasoning (Huang et al., 2023).

### 2.2.2 FURTHER DISCUSSIONS ABOUT THE THEORETICAL RESULTS

According to the propositions above, in both CoT-SC and MAD, the bounds of the probability that the answer to the given query is correctly inferred are discussed in two different cases. Specifically, when $p < \frac{1}{2}$ or $e^{-\alpha N_\mathrm{e}} < \frac{1}{2}$, the probability is upper bounded, while the probability is lower bounded when $p \geq \frac{1}{2}$ or $e^{-\alpha N_\mathrm{e}} \geq \frac{1}{2}$. The two cases implicitly correspond to two types of reasoning cases: *hard problem reasoning* and *easy problem reasoning*. Detailed discussions are provided as follows.

**Hard Problem Reasoning.** In the context of hard problem reasoning (HPR) setting (i.e., $p < \frac{1}{2}$ or $e^{-\alpha N_\mathrm{e}} < \frac{1}{2}$), it assumes that LLM agents can hardly infer the correct answer to the query due to the difficulty of the problem and many erroneous memories in the previous debate round. In CoT-SC, when the capability of LLM agents (i.e., the probability $p$) is fixed, increasing the number of responses deteriorates the performance exponentially. In contrast, when the number of responses (i.e., $N_\mathrm{sc}$) is fixed, employing more powerful LLM agents ($p \to \frac{1}{2}$) tends to improve the reasoning performance. As a comparison, in MAD, the performance is mainly determined by the number of agents $N_\mathrm{a}$ and erroneous memories $N_\mathrm{e}$. In the case of a 2-round MAD, on the one hand, when the number of agents increases (with $N_\mathrm{e}$ fixed), the performance deteriorates exponentially; on the other hand, with $N_\mathrm{a}$ fixed, when the number of erroneous memories decreases (i.e., $e^{-\alpha N_\mathrm{e}} \to \frac{1}{2}$), the performance of MAD is improved and approaches to the performance of CoT-SC. Thus, the performance of CoT-SC can be viewed as an upper bound of that of MAD in HPR. Both aspects above indicate that increasing the number of samples (i.e., $N_\mathrm{sc}$ and $N_\mathrm{a}$) in the HPR setting tends to result in performance collapse.

**Easy Problem Reasoning.** Different from the hard problem reasoning setting, LLM agents can easily infer the answer of the query in the context of the easy problem reasoning setting (EPR, i.e., $p \geq \frac{1}{2}$ or $e^{-\alpha N_\mathrm{e}} \geq \frac{1}{2}$). In this case, for both CoT-SC and MAD frameworks, on the one hand, increasing the number of sampled answers or agents (i.e., $N_\mathrm{sc}$ or $N_\mathrm{a}$) helps improve the performance. On the other hand, employing powerful LLM agents (i.e., $p \to 1$) or reducing erroneous memories (i.e., $e^{\alpha(j-N_\mathrm{a})} \to 1$) also contributes to improving the performance of the two paradigms in reasoning.

Thus, with the discussions above, we can summarize two insights. On the one hand, simply increasing sampling numbers (i.e., $N_\mathrm{a}$) does not improve the performance in all cases, while reducing erroneous memories (i.e., $N_\mathrm{e}$) consistently improves the performance of MAD. On the other hand, the performance of MAD is also constrained by the reasoning performance in the previous debate round (i.e., $\omega_j$), thereby being theoretically bounded by CoT-SC. With these two observations taken into consideration, we propose to improve the performance of MAD by removing the erroneous memories to remove the constraint above and guarantee the performance of both HPR and EPR being improved.

## 3 MAD-M$^2$: MULTI-AGENT DEBATE WITH MEMORY MASKING

In this section, we propose a simple yet effective multi-agent framework, *multi-agent debate with memory masking* (MAD-M$^2$), to critically evaluate memories from the last debate round and mask erroneous memories so that reasoning can be performed with correct memories in the next debate.

### 3.1 DETAILED IMPLEMENTATION OF MAD-M$^2$

An overview of our proposed MAD-M$^2$ framework is visualized in Fig. 2. In general, the core idea of our proposed MAD-M$^2$ framework is inserting critical evaluation and masking operations between two debate rounds to identify and remove the potential erroneous memories for the next debate round.

**Step 1. Initial Debate Round.** In the MAD-M$^2$ framework, at the initial debate round, all $N_\mathrm{a}$ LLM agents are required to independently generate responses to the given query $x^\mathrm{test} \in \mathcal{X}$ without any contextual information. Specifically, a total of $N_\mathrm{a}$ responses are independently generated from LLM agents $A_{\theta_i} \in \mathcal{A}$, respectively, and formulate the memory vector of the initial round: $\mathcal{M}_1 =$

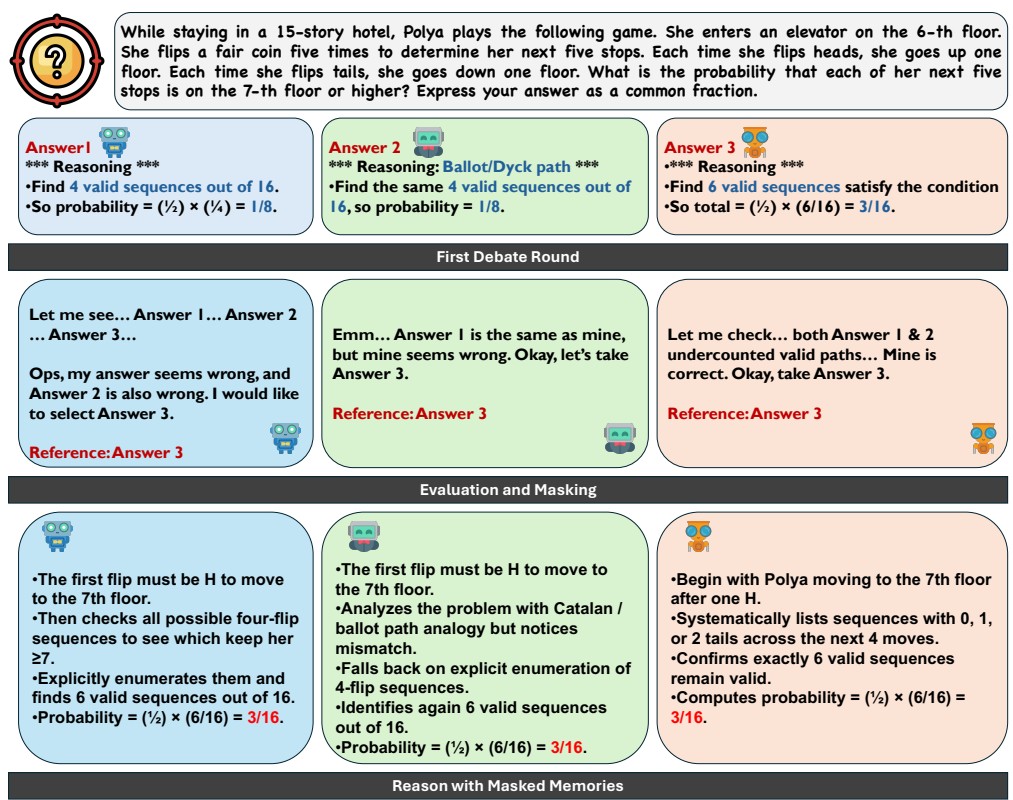

Figure 2: **An illustration of MAD-M$^2$ framework.** In general, MAD-M$^2$ mainly includes three steps. (i) In the initial debate round, LLM agents independently generate responses based on the given query. (ii) The responses generated in the previous round are treated as memories. All memories will be critically evaluated and the potential erroneous memories will be masked for the reasoning in the next debate round. (iii) With the preserved memories, agents perform reasoning in the next debate.

$[A_{\theta_1}(x^{\text{test}}, \emptyset), A'_{\theta_2}(x^{\text{test}}, \emptyset), ..., A_{\theta_{N_a}}(x^{\text{test}}, \emptyset)]$, where $A'_{\theta_i}(x^{\text{test}}, \emptyset)$ denotes the wrong reasoning responses generated by the $i$-th LLM agent while $A_{\theta_i}(x^{\text{test}}, \emptyset)$ denotes correct reasoning responses.

**Step 2. Evaluation and Masking.** After a round of debate, a set of memories regarding the given query is obtained. Here, we consider a case where both correct and incorrect reasoning responses are contained. Specifically, the memories can be completely correct, partially correct, or completely wrong. According to our previous demonstration, in the case where erroneous memories are included, the capability of agents generating correct responses in the next debate round will be undermined. To solve this problem, in MAD-M$^2$, we propose to first critically evaluate the memories from the previous debate and then generate a binary vector $M = \{0, 1\}^{N_a}$ to mask the potential erroneous memories identified by the agent. Then, we obtain a set of memories with erroneous ones removed:

$$\widehat{\mathcal{M}}_r^{(i)} = M^{(i)} \odot \mathcal{M}_r, \text{where } M^{(i)} = g_{A_{\theta_i}}^{\text{map}}(\mathcal{M}_r), \tag{2}$$

where $g_{A_{\theta_i}}^{\text{map}} : \mathcal{M}_r \rightarrow \{0, 1\}^{N_a}$ denotes an operation that maps the evaluation of the agent $A_{\theta_i}$ on the $r$-th debate memory $\mathcal{M}_r$ into a binary mask vector, $\widehat{\mathcal{M}}_r^{(i)}$ denotes the new set of memories selected by the agent from the memory, and $\odot$ denotes the element-wise multiplication. Since all agents are initialized from the same model, the masking operation can be simply performed once for all agents.

**Subjective Masking Strategy.** In the subjective masking strategy, memories are masked according to the subjective evaluations of LLM agents. Specifically, in MAD-M$^2$, LLM agents are required to evaluate memories with the flags "YES", "NO", and "NOT SURE". Depending on the strictness of the predefined filtering rule, the "NOT SURE" is treated as either "YES" or "NO" correspondingly.

**Objective Masking Strategy.** Inspired by previous work (Fu et al., 2025), we also leverage the perplexity of LLMs to perform memory masking. Since high perplexity usually implies that LLMs are not confident of the generated content, answers with high perplexity may probably contain erroneous

information (e.g., fallacious content and hallucinations). Thus, as an objective masking strategy, the perplexity of responses is measured, and only the response with the lowest perplexity is preserved.

**Step 3. Reasoning with Masked Memories.** With the new set of retained memories $\widehat{\mathcal{M}}_r^{(i)}$ obtained as contextual information through Eq. (2), in the $r+1$-th debate round, a set of new reasoning responses $\mathcal{M}_{r+1} = [A'_{\theta_1}(x^{\text{test}}, \widehat{\mathcal{M}}_r^{(1)}), A_{\theta_2}(x^{\text{test}}, \widehat{\mathcal{M}}_r^{(2)}), ..., A_{\theta_{N_a}}(x^{\text{test}}, \widehat{\mathcal{M}}_r^{(N_a)})]$, where both correct and incorrect responses are included, is generated from agents with the query and the refined memories.

In our proposed MAD-M$^2$ framework, steps 2 and 3 are conducted iteratively until the end of the debate. At the final round of debate, majority voting is performed to obtain the answer to the query.

## 3.2 MORE DISCUSSIONS

In the previous section, we introduce a simple yet effective MAD-M$^2$ method to improve the robustness of the conventional multi-agent debate framework by removing the potential incorrect memories derived in the last debate round. In this section, we propose to provide more analyses and discussions about the MAD-M$^2$ framework to have a comprehensive understanding of its property.

**Token Consumption Analysis.** Due to the multi-round interactions among agents in the multi-agent debate paradigm, the consumption of tokens of reasoning paradigms has become a significant concern in recent works (Liu et al., 2024; Zeng et al., 2025). In our proposed subjective masking strategy, the introduction of the self-evaluation step tends to result in more token consumptions. Thus, in order to quantitatively compare naive MAD and our proposed MAD-M$^2$ frameworks, we follow Liu et al. (2024) to conduct a detailed analysis on the token consumption for both naive MAD and MAD-M$^2$.

Let's denote the token of the query $x^{\text{test}}$ as $T^{\text{q}}$, the token of the output of the agent $A_{\theta_i}$ at the $r$-th debate round as $T^{\text{o}}_{i,r}$. Then, we formulate the token consumption of both MAD and MAD-M$^2$ as:

$$N_{\text{MAD}}^{\text{token}} = N_a N_{\text{round}} T^{\text{q}} + N_a \sum_{r=2}^{N_{\text{round}}} \sum_{i=1}^{N_a} T^{\text{o}}_{r-1,i} + \sum_{r=1}^{N_{\text{round}}} \sum_{i=1}^{N_a} T^{\text{o}}_{r,i},$$

$$N_{\text{MAD-M}^2}^{\text{token}} \leq N_a N_{\text{round}} T^{\text{q}} + 2N_a \sum_{r=2}^{N_{\text{round}}} \sum_{i=1}^{N_a} T^{\text{o}}_{r-1,i} + \sum_{r=1}^{N_{\text{round}}} \sum_{i=1}^{N_a} T^{\text{o}}_{r,i}.$$

According to the results above, compared to the conventional MAD framework, our proposed MAD-M$^2$ consumes more tokens. Specifically, even in the worst case, where all memories are preserved, MAD-M$^2$ consumes $N_a \sum_{r=2}^{N_{\text{round}}} \sum_{i=1}^{N_a} T^{\text{o}}_{r-1,i}$ more input tokens than the naive MAD framework.

**Comparison to Existing Works.** In literature, many works have explored memory selection in the multi-agent debate framework (Liu et al., 2024; Li et al., 2024; Zeng et al., 2025). Specifically, Li et al. (2024) proposes Sparse MAD (S-MAD) to organize agents in MAD as a graph. One agent can access the response of the other only if the two agents are connected. Moreover, Zeng et al. (2025) proposes Selective Sparse MAD (S$^2$-MAD) to reduce the exchange of trivial memories and unproductive discussions among agents. Among these works, the most related work to our proposed MAD-M$^2$ is S-MAD. However, S-MAD only considers static (predefined) topologies for sparse communications among agents. Although dynamic topology is also mentioned in the paper, the memories are selected in a random way. However, in our proposed MAD-M$^2$ framework, the memories selected from the last debate round and the sparsity of communications are dynamically determined by LLM agents with a cost of more token consumption or the internal states (e.g., perplexity) regarding the memories.

## 4 EXPERIMENTS

In this section, we evaluate our proposed MAD-M$^2$ on both mathematical reasoning and language understanding benchmarks. We first briefly introduce the experimental settings. Then, we provide detailed quantitative results and further analyses to validate the efficacy and obtain a comprehensive understanding of MAD-M$^2$. Complete implementations and settings are available in Appendix D.

### 4.1 EXPERIMENTAL SETUPS

**Models.** In our experiments, we mainly consider evaluating MAD-M$^2$ equipped with four mainstream open-source large language models: Qwen2.5-7B-Instruct (Yang et al., 2024a), Qwen2.5-Math-7B-Instruct (Yang et al., 2024b), DeepSeek-Math-7B-Instruct (Shao et al., 2024), and QwQ-32B (Team, 2025). Compared to Qwen2.5-7B-Instruct, other models are more powerful in math reasoning tasks.

Table 1: Empirical results of accuracy (with standard deviation) and token consumption ($T$.). We evaluate four mainstream open-source LLMs on both mathematical reasoning and language understanding benchmarks. We highlight the best performance in **bold**, and the second-best performance in underline. For fairness, all results are the average of five trials on different seeds (i.e., 41-45).

| Methods | AIME24 | | AIME25 | | MMLU_Pro | | MATH | | GSM8K | |
|---|---|---|---|---|---|---|---|---|---|---|
| | Acc. (%)↑ | $T.↓$ | Acc. (%)↑ | $T.↓$ | Acc. (%)↑ | $T.↓$ | Acc. (%)↑ | $T.↓$ | Acc. (%)↑ | $T.↓$ |
| Qwen2.5-7B-Instruct | | | | | | | | | | |
| CoT | 3.3 | ×0.07 | 3.3 | ×0.08 | 36.0±6.3 | ×0.08 | 49.2±3.8 | ×0.07 | 64.8±5.4 | ×0.09 |
| CoT-SC | 10.0 | ×0.43 | **10.0** | ×0.45 | 39.2±4.0 | ×0.51 | **58.0±1.0** | ×0.45 | 83.6±3.9 | ×0.51 |
| MAD | **13.3** | ×1.00 | 6.7 | ×1.00 | 43.0±2.2 | ×1.00 | 55.6±3.7 | ×1.00 | **91.8±2.4** | ×1.00 |
| MAD-M²(S) | **13.3** | ×1.13 | 3.3 | ×1.21 | **43.6±2.3** | ×1.17 | 56.8±2.1 | ×1.20 | 89.0±4.0 | ×1.25 |
| MAD-M²(O) | 6.7 | ×0.68 | 6.7 | ×0.65 | 42.4±5.3 | ×0.69 | 54.2±2.6 | ×0.67 | 89.0±2.0 | ×0.72 |
| Qwen2.5-Math-7B-Instruct | | | | | | | | | | |
| CoT | 13.3 | ×0.08 | 10.0 | ×0.08 | 39.6±0.9 | ×0.08 | 77.8±5.2 | ×0.08 | 95.2±1.6 | ×0.08 |
| CoT-SC | **23.3** | ×0.53 | 10.0 | ×0.48 | **41.4±5.1** | ×0.47 | **82.0±4.7** | ×0.44 | **96.4±1.7** | ×0.45 |
| MAD | 6.7 | ×1.00 | 6.7 | ×1.00 | 34.2±2.9 | ×1.00 | 71.2±3.3 | ×1.00 | 95.2±1.8 | ×1.00 |
| MAD-M²(S) | 6.7 | ×1.37 | 6.7 | ×1.37 | 35.0±2.2 | ×1.36 | 71.2±3.3 | ×1.41 | 95.2±1.8 | ×1.44 |
| MAD-M²(O) | 13.3 | ×0.67 | **13.3** | ×0.62 | 37.0±2.9 | ×0.62 | 80.2±3.8 | ×0.62 | 95.4±1.7 | ×0.60 |
| DeepSeek-Math-7B-Instruct | | | | | | | | | | |
| CoT | 0.0 | ×0.07 | 0.0 | ×0.09 | 27.8±7.9 | ×0.17 | 34.2±4.5 | ×0.08 | 79.0±3.8 | ×0.09 |
| CoT-SC | **3.3** | ×0.44 | 0.0 | ×0.46 | **32.2±4.3** | ×0.99 | **44.4±3.9** | ×0.47 | **88.8±2.6** | ×0.52 |
| MAD | 0.0 | ×1.00 | 0.0 | ×1.00 | 31.2±5.4 | ×1.00 | 38.6±2.6 | ×1.00 | 81.2±2.7 | ×1.00 |
| MAD-M²(S) | 0.0 | ×1.32 | 0.0 | ×1.30 | 30.8±5.2 | ×1.66 | 37.0±5.1 | ×1.31 | 80.8±3.5 | ×1.33 |
| MAD-M²(O) | 0.0 | ×0.67 | 0.0 | ×0.67 | 30.8±6.4 | ×0.75 | 39.8±3.6 | ×0.68 | 82.2±4.4 | ×0.71 |
| QwQ-32B | | | | | | | | | | |
| CoT | **80.0** | ×0.13 | 56.7 | ×0.14 | 75.2±4.9 | ×0.11 | 80.8±1.6 | ×0.10 | 97.4±2.3 | ×0.08 |
| CoT-SC | **80.0** | ×0.85 | **80.0** | ×0.85 | **76.4±6.8** | ×0.63 | **81.6±0.9** | ×0.61 | 97.4±2.3 | ×0.44 |
| MAD | 76.7 | ×1.00 | 73.3 | ×1.00 | 75.4±4.2 | ×1.00 | 79.2±2.8 | ×1.00 | 97.2±2.3 | ×1.00 |
| MAD-M²(S) | 76.7 | ×1.28 | 73.3 | ×1.25 | 75.8±6.3 | ×1.22 | 79.6±2.3 | ×1.27 | **97.8±1.9** | ×1.32 |
| MAD-M²(O) | **80.0** | ×0.67 | 76.7 | ×0.90 | 75.2±5.9 | ×0.69 | 75.0±3.9 | ×0.67 | 96.6±1.8 | ×0.56 |

(a) Qwen2.5-7B (S)  (b) Qwen2.5-7B (L)  (c) Qwen2.5-Math-7B (S)  (d) Qwen2.5-Math-7B (L)

Figure 3: **Visualization of erroneous memory identification of different LLMs.** We here examine the erroneous memory identification capability of different LLMs. "S" denotes the strict rule, and "L" denotes the loose rule. According to the results, the objective masking strategy generally works better on the powerful LLMs, while the subjective masking works better on relatively weak LLMs.

**Benchmarks.** In our experiments, we evaluate MAD-M² on both mathematical reasoning benchmarks (i.e., GSM8K (Cobbe et al., 2021), MATH (Hendrycks et al., 2021), AIME24&25) and language understanding benchmarks (i.e., MMLU_Pro (Wang et al., 2024)). In this case, GSM8K, MATH and MMLU_Pro are treated as easy reasoning tasks, while AIME24&25 are the hard reasoning tasks.

**Baselines.** In our experiments, the following frameworks are adopted as baselines: (1) Chain-of-Thought (CoT) (Wei et al., 2022b); (2) Self-Consistency Chain-of-Thoughts (CoT-SC) (Wang et al., 2023) with 6 independent reasoning paths; and (3) Multi-Agent Debate (MAD) (Du et al., 2023). For all MAD frameworks above, the number of agents and debate rounds are set to 3 and 2, respectively.

## 4.2 MAIN RESULTS

We evaluate MAD-M² with both subjective (MAD-M²(S)) and objective (MAD-M²(O)) masking strategies on both mathematical reasoning and language understanding benchmarks. The quantitative results, including accuracy (with standard deviation) and token consumption, are reported in Table 1.

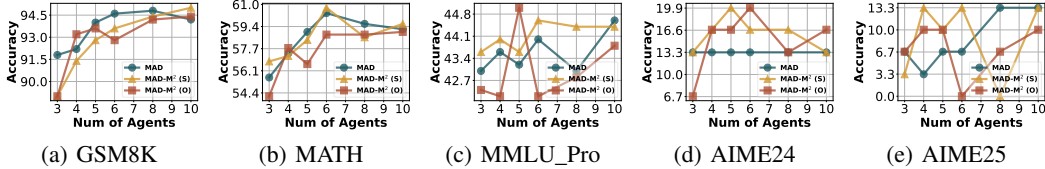

(a) GSM8K     (b) MATH     (c) MMLU_Pro     (d) AIME24     (e) AIME25

Figure 4: **Effect of scaling the number of agents in the case of Qwen2.5-7B-Instruct.** The number of agents is increased from 3 to 10. According to the figures, both frameworks benefit from the increase of the number of agents and MAD-M$^2$(S) tends to achieve better performance in most cases.

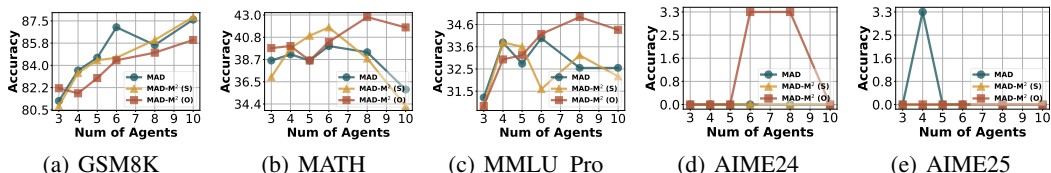

(a) GSM8K     (b) MATH     (c) MMLU_Pro     (d) AIME24     (e) AIME25

Figure 5: **Effect of scaling the number of agents in the case of DeepSeek-Math-7B-Instruct.** The number of agents is increased from 3 to 10. According to the figures, both frameworks act differently when the number of agents increases and MAD-M$^2$(O) achieves better performance in most cases.

*Observation 1.* **MAD-M$^2$outperforms MAD in most cases.** As shown in Table 1, we can observe that MAD-M$^2$ achieves better performance than MAD in most cases with different LLMs. Specifically, in the case of Qwen2.5-7B-Instruct, MAD-M$^2$(S) achieves 0.6% and 1.2% improvements on MMLU_Pro and MATH benchmarks, respectively. Moreover, in the case of Qwen2.5-Math-7B-Instruct, MAD-M$^2$(O) achieves 2.8%, 9.0%, and 0.2% on MMLU_Pro, MATH, and GSM8K benchmarks. Meanwhile, MAD-M$^2$(O) also achieves 6.6% improvements on both AIME24&25.

*Observation 2.* **MAD with powerful LLMs tends to achieve better performance on reasoning tasks.** In Table 1, MAD with weak LLMs, such as Qwen2.5-7B-Instruct, achieves better performance on easier tasks (e.g., MATH and MMLU_Pro), while MAD with powerful LLMs, such as Qwen2.5-Math-7B and QwQ-32B, achieves better performance on both easy and hard benchmarks simultaneously. This phenomenon aligns with our theoretical analyses that MAD benefits from the powerful reasoning capability. Masking erroneous memories helps improve the reasoning capability.

*Observation 3.* **The efficacy of masking strategies is highly dependent on the intrinsic capability of LLMs.** As shown in Table 1, in the case of MAD with weak LLMs (e.g., Qwen2.5-7B-Instruct), MAD-M$^2$(S) generally achieves better performance than MAD-M$^2$(O). However, in the case of MAD with powerful LLMs (e.g., Qwen2.5-Math-7B-Instruct & QwQ-32B), MAD-M$^2$(O) demonstrates its advantage on more challenging AIME benchmarks. Specifically, MAD-M$^2$(O) achieves significantly better performance than MAD-M$^2$(S) with about 50% tokens. This indicates that the perplexity-based masking strategy provides a more precise signal for erroneous memory identification in reasoning.

### 4.3 MORE ANALYSES

**Token Consumption.** We here take the conventional MAD framework as the baseline and compare the token consumptions of all reasoning frameworks in Table 1. According to the results, multi-agent reasoning paradigms typically consume more tokens than single-agent paradigms (i.e., CoT & CoT-SC) due to multi-round interactions (i.e., inputs and outputs) and memories in prompts. Consistent with the analysis results in Section 3.2, subjective masking consumes more tokens (13%~41%) due to the extra subjective evaluation phase. In contrast, when integrating objective masking with MAD-M$^2$, the token consumption decreases by about 30% in almost all cases due to fewer memories in prompts.

**Erroneous Memory Identification.** To figure out whether MAD-M$^2$ can correctly identify those erroneous memories, we consider two different rules here: (1) *Strict rule*: All erroneous memories are identified. (2) *Loose rule*: Correct memories are dominant in the retained memories. The results are visualized in Fig. 3. Note that the strict rule does not perfectly fit objective masking since only one answer is preserved when performing objective masking. According to the results, on the one hand, we can observe that MAD-M$^2$ consistently achieves good performance on simple reasoning tasks (e.g., GSM8K & MATH). On the other hand, subjective masking generally achieves better performance with the loose rule, where correct memories take a dominant role in preserved memories. In addition, MAD-M$^2$ with weak LLMs prefers subjective masking (cf. Figs. 3(a) & 3(b)) while

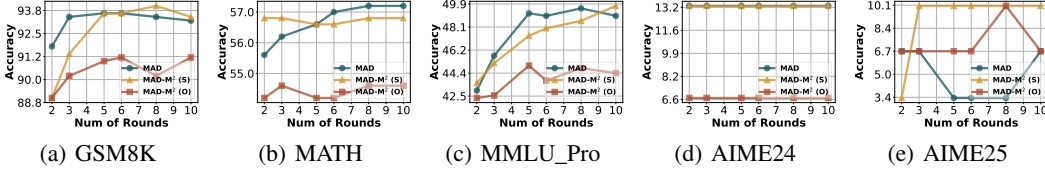

| (a) GSM8K | (b) MATH | (c) MMLU_Pro | (d) AIME24 | (e) AIME25 |

Figure 6: **Effect of scaling the number of debate rounds in the case of Qwen2.5-7B-Instruct.** The number of debate rounds increases from 2 to 10. According to the figures, both frameworks basically benefit from the increase in debate rounds. In most cases, MAD-M$^2$(S) outperforms MAD-M$^2$(O).

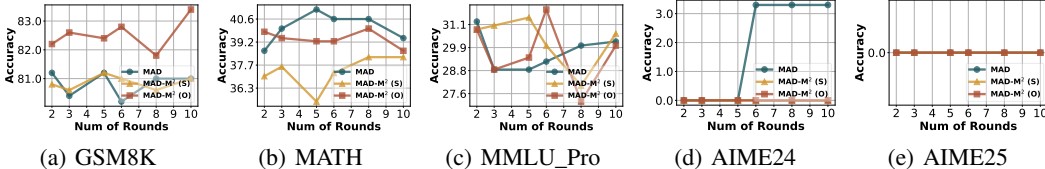

| (a) GSM8K | (b) MATH | (c) MMLU_Pro | (d) AIME24 | (e) AIME25 |

Figure 7: **Effect of scaling the number of debate rounds w.r.t. DeepSeek-Math-7B-Instruct.** The number of debate rounds increases from 2 to 10. According to the figures, the performance of different cases varies. In most cases, MAD-M$^2$(O) achieves better performance than MAD-M$^2$(S).

MAD-M$^2$ with powerful LLMs prefers objective masking, in particular on hard reasoning tasks (cf. Figs. 3(c) & 3(d)). Generally, objective masking is more robust as the difficulty of tasks increases.

**Scaling of Number of Agents & Debate Rounds.** We investigate the scaling capability of MAD-M$^2$. As a comparison, we take MAD as the baseline. In the case of scaling the number of agents, we fix the number of debate rounds as 2 and increase the number of agents to 4, 5, 6, 8, and 10, respectively. The results of both MAD and MAD-M$^2$ equipped with two different LLMs (Qwen2.5-7B and DeepSeek-Math-7B) are visualized in Figs. 4 and 5. Empirically, both MAD and MAD-M$^2$ benefit from the increase in the number of agents. From the perspective of masking strategy, subjective masking performs better than objective masking on MAD-M$^2$ with weak LLMs (e.g., Qwen2.5-7B-Instruct) while objective masking works better on powerful LLMs (e.g., Qwen2.5-Math-7B-Instruct). In the case of scaling the number of debate rounds, we fix the number of agents to 3 and increase the number of debate rounds to 4, 5, 6, 8, and 10, respectively. The results with respect to two LLMs are visualized in Figs. 6 and 7. Increasing the number of debate rounds does not consistently improve performance in all cases. Specifically, increasing the number of debate rounds benefits the frameworks with Qwen2.5-7B-Instruct. However, in the case where the powerful models (e.g., DeepSeek-Math-7B-Instruct) are equipped, with the increase of debate rounds, the performance even drops. In addition, consistent with the case of scaling the number of agents, subjective masking performs better than objective masking on MAD-M$^2$ with weak LLMs while objective masking works better on MAD-M$^2$ with powerful LLMs. More analysis results are available in Appendix E.1

**Comparison between MAD-M$^2$ and Sparse MAD.** We here compare the performance between our proposed MAD-M$^2$ and Sparse MAD, which is a related work of our method. Here, we consider an MAD with 6 agents and 2 debate rounds. For Sparse MAD, each agent only obtains memories from the two agents it connects to. The results are reported in Table 3. Generally, MAD-M$^2$ achieves better performance than Sparse MAD in most cases, especially on those hard reasoning tasks (e.g., AIME24 & 25). For example, in the case of Qwen2.5-7B-Instruct, MAD-M$^2$ outperforms Sparse MAD. Although Sparse MAD excels MAD-M$^2$ on MATH in the case of Qwen2.5-Math-7B-Instruct, its catastrophic drop in MMLU_Pro suggests a lack of robust generalization compared to our proposed MAD-M$^2$. Thus, MAD-M$^2$ can generally achieve more balanced performance on the reasoning tasks.

## 5 CONCLUSION

Our work mainly focuses on the robustness of the conventional multi-agent debate framework. In this paper, we find that MAD is vulnerable to erroneous memories from the last debate round. The erroneous memories may mislead agents from the original correct reasoning to erroneous ones. To understand this phenomenon, we theoretically demonstrate that the reasoning capability of LLM agents in the next debate round is closely related to the number of erroneous memories in the previous round. Inspired by this observation, we propose a simple yet effective framework, multi-agent debate with memory masking, to enhance the reasoning capability of MAD by masking potential erroneous memories. Extensive experiments and analyses demonstrate the efficacy of our proposed method.

ACKNOWLEDGEMENT

HDT, XF and BH were supported by RGC Young Collaborative Research Grant No. C2005-24Y, Tencent WeChat Faculty Research Award, and HKBU CSD Departmental Incentive Scheme.

ETHICS STATEMENT

We believe that our work raises no significant ethical concerns.

REPRODUCIBILITY STATEMENT

In this paper, both theoretical and empirical results are included. For theoretical results, the necessary assumptions and the completed proofs have been provided in the main paper and the appendix, respectively. For the reproducibility of our experiments, we have provided detailed explanations about our experimental settings and the necessary introduction to the datasets in our paper.

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

## A  DETAILED RELATED WORK

Recently, more and more attention has been attracted to large language models (Achiam et al., 2023; Yang et al., 2024a), which are parameterized with millions of weights and trained on numerous corpora. Along with the impressive improvements in performance on conventional natural language tasks, one important emergent ability of these large language models is performing reasoning tasks.

**LLM Reasoning.** As the most impressive emergent capability of large language models, reasoning has received more and more attention from the community. Such a capability enables LLMs to address complex questions in a logical way as human beings based on their prior knowledge without further training. So far, numerous research works have been conducted to improve the reasoning capability of LLMs. One typical framework to enhance the reasoning capability of LLMs is Chain-of-Thought (CoT) (Wei et al., 2022b). In CoT, LLMs are instructed to break the tasks into sequential steps and solve the problem step by step. Based on CoT, some other variants are proposed to further improve the reasoning capability of LLMs (Zelikman et al., 2022; Wang et al., 2023; Shum et al., 2023). Among these works, a famous work is Self-Consistency Chain-of-Thought (CoT-SC) (Wang et al., 2023). CoT-SC improves the reasoning capability of LLMs significantly by simply sampling multiple reasoning trajectories and performing majority voting on these trajectories for the final answer. In addition to modeling the reasoning trajectories into the chain of thoughts, other works also explore other logical structures. For example, Jin & Lu (2023) (Tab-CoT) models the reasoning trajectories in a highly structured way. Yao et al. (2023) models the reasoning trajectories into a tree-like path (a.k.a. Tree-of-Thought, ToT). This further facilitates evaluating multiple reasoning paths and self-assessment. However, both chain-based and tree-based reasoning paradigms are constrained to address linear reasoning tasks. To solve this problem, Graph-of-Thoughts (Besta et al., 2024) is proposed to model the reasoning trajectory into a flexible graph to solve non-linear tasks.

**Multi-agent Debate.** Multi-agent debate (MAD) (Du et al., 2023) is equivalent to a scaling of conventional reasoning paradigms. By comparing the answers derived from a set of LLMs, the fallacious content can be removed, and the correct answers can be achieved. The main goal of MAD is to perform reasoning with multiple LLMs in the way of debate as human beings do. Specifically, in this case, the reasoning is performed by allowing multiple LLMs to debate on their previous responses in a multi-round way. The final answer is obtained via performing majority voting on the set of responses in the last debate round. By considering previous responses, the answers generated in the new round can be further refined. Compared to the conventional reasoning paradigms, such as CoT, MAD consumes more resources (e.g., tokens and time) (Du et al., 2023; Li et al., 2024; Liu et al., 2024; Zeng et al., 2025). Moreover, due to the randomness of response generation, multi-agent debate also suffers from the noisy responses (Zeng et al., 2025). Thus, a series of works is proposed to reduce the resource consumption and improve the performance of MAD by polishing the memories. Specifically, Sparse MAD (S-MAD) (Li et al., 2024) proposes to formulate the MAD framework as a graph and sparsify the topology of the graph, which is able to simultaneously reduce resource consumption and improve the performance. Moreover, Liu et al. (2024) proposes Group Debate (GD) to divide all agents into several debate groups and only allow agents to debate in their own group. Further, Selective Sparse MAD ($S^2$-MAD (Zeng et al., 2025)) is proposed to reduce the redundant content and unproductive discussions in debate to improve the efficiency and performance of MAD. Recently, Choi et al. (2025) conducted a theoretical analysis on the MAD framework, demonstrating that majority voting plays an important role in MAD achieving good performance in reasoning tasks.

**Context Engineering.** Context Engineering methods (Mei et al., 2025) manipulate unstructured streaming LLM reasoning to construct structured context systems, enabling sophisticated LLM-driven MAD systems and agentic systems by managing scaling context length, where numerous tokens are irrelevant or confusing for answers. Recent advancements in context engineering are two-fold: context augmentation and context compression. Context augmentation methods aim at extending critical information in context to optimize reasoning performance. Retrieval-augmented Generation (RAG) methods (Lewis et al., 2020; Gao et al., 2023; Fan et al., 2024) provide access to external information sources, including databases (Lian et al., 2024), knowledge graphs (Sun et al., 2024; LUO et al., 2024), and textual document collections (Asai et al., 2024; Sarthi et al., 2024), enabling access to relevant knowledge. Additionally, comprehensive feedback approaches, such as Self-Refine (Madaan et al., 2023), employ LLMs to evaluate their prior reasoning for self-improvement, while Reflexion (Shinn et al., 2023), STaR (Zelikman et al., 2022), and AlphaEvolve (Novikov et al., 2025) introduce reliable external environment feedback for consistent performance improvement. Different from

context augmentation, context compression methods aim at addressing the overlong context issue that challenges the finite context window size by dynamically reducing the contextual information. For example, INFTYTHINK (Yan et al., 2025) iteratively summarizes prior reasoning contexts for short-context reasoning, while CoThink (Fan et al., 2025) employs short-response LLMs to orchestrate reasoning strategy and dynamically control context length. Additionally, O1-Pruner (Luo et al., 2025), L1 (Aggarwal & Welleck, 2025), and DAST (Shen et al., 2025) consider the reasoning length during the training phase to encourage LLMs to address the given queries with fewer tokens.

## B PROOFS

### B.1 PROOF OF PROPOSITION 2.2

*Proof.* Since CoT-SC performs reasoning on the given query by voting the majority on a set of independently generated multiple responses, given Assumption 2.1, the number of the correct responses conforms to a binomial distribution $N_{\text{cor}} \sim \mathcal{B}(N_{\text{sc}}, p)$:

$$P(N_{\text{cor}} = k) = \binom{N_{\text{sc}}}{k} p^k (1-p)^{N_{\text{sc}}-k},$$

where $\mathcal{B}$ denotes the binomial distribution. Thus, the case that CoT-SC correctly answers the question can be formulated as the event: $N_{\text{cor}} > \frac{N_{\text{sc}}}{2}$. Then, we further have the probability that CoT-SC correctly infers the answer to the query

$$P(N_{\text{cor}} > \frac{N_{\text{sc}}}{2}) = \sum_{k=\lfloor \frac{N_{\text{sc}}}{2} \rfloor+1}^{N_{\text{sc}}} P(N_{\text{cor}} = k)$$

$$= \sum_{k=\lfloor \frac{N_{\text{sc}}}{2} \rfloor+1}^{N_{\text{sc}}} \binom{N_{\text{sc}}}{k} p^k (1-p)^{N_{\text{sc}}-k},$$

where $\lfloor \cdot \rfloor$ is the flooring operator, and $\binom{N_{\text{sc}}}{k} = \frac{N_{\text{sc}}!}{k!(N_{\text{sc}}-k)!}$ is the binomial coefficient with $N_{\text{sc}}$ and $k$.

Consider a set of independent random variables $\{X_i\}_{i=1}^{N_{\text{sc}}}$, where $X_i \in \{0,1\}$, $P(X_i = 1) = p$ and $P(X_i = 0) = 1 - p$. Let $N_{\text{cor}} = \sum_{i=1}^{N_{\text{sc}}} X_i$, we then have $E[X_i] = p$ and $E[N_{\text{cor}}] = N_{\text{sc}} p$. With the Hoeffding Inequality, we have

$$P(N_{\text{cor}} - E[N_{\text{cor}}] \geq t) \leq \exp\left(-\frac{2t^2}{N_{\text{sc}}}\right),$$

If $p < \frac{1}{2}$, then $E[N_{\text{cor}}] = N_{\text{sc}} p < \frac{N_{\text{sc}}}{2}$. Let $t = (\frac{1}{2} - p)N_{\text{sc}} > 0$:

$$P(N_{\text{cor}} > \frac{N_{\text{sc}}}{2}) \leq \exp\left(-2N_{\text{sc}}\left(\frac{1}{2} - p\right)^2\right).$$

Else, if $p \geq \frac{1}{2}$, then $E[N_{\text{cor}}] \geq \frac{N_{\text{sc}}}{2}$. Thus, we consider the complementary case: $P(N_{\text{cor}} - E[N_{\text{cor}}] \leq \frac{N_{\text{sc}}}{2})$. Then, let $t = (p - \frac{1}{2})N_{\text{sc}} > 0$, we can get:

$$P(N_{\text{cor}} \leq \frac{N_{\text{sc}}}{2}) \leq \exp\left(-2N_{\text{sc}}\left(\frac{1}{2} - p\right)^2\right).$$

Thus, for the original case $P(N_{\text{cor}} > \frac{N_{\text{sc}}}{2})$, we have

$$P(N_{\text{cor}} > \frac{N_{\text{sc}}}{2}) \geq 1 - \exp\left(-2N_{\text{sc}}\left(\frac{1}{2} - p\right)^2\right).$$

The proof is completed. □

### B.2 PROOF OF PROPOSITION 2.3

*Proof.* In a 2-round multi-agent debate case, where $N_{\mathrm{a}}$ agents are involved in each debate round, we first consider the initial debate round, where the responses are only conditioned on the query prompt. Thus, the probability of MAD achieving $N_{\mathrm{cor}}^{(1)} \in \{0, 1, 2, ..., N_{\mathrm{a}}\}$ correct reasoning responses is:

$$P(N_{\mathrm{cor}}^{(1)} = j) = \binom{N_{\mathrm{a}}}{j} p^j (1-p)^{N_{\mathrm{a}}-j}, \text{ where } j \in \{0, 1, 2, ..., N_{\mathrm{a}}\}.$$

For the second round, assume that the probability of LLM agents correctly inferring the answer is modified to $e^{-\alpha N_{\mathrm{e}}}$, where $N_{\mathrm{e}} = N_{\mathrm{a}} - N_{\mathrm{cor}}^{(1)}$, depending on the number of correct memories in the previous debate round. Note that new responses are generated independently in the second round among agents, although the generated answers are conditioned on memories, and the number of correct responses in the second round also conforms to a binomial distribution $N_{\mathrm{cor}}^{(2)} \sim \mathcal{B}(N_{\mathrm{a}}, e^{\alpha(j-N_{\mathrm{a}})})$, $\alpha > 0$. In this case, the probability that the answer is correctly inferred is:

$$P(N_{\mathrm{cor}}^{(2)} > \frac{N_{\mathrm{a}}}{2}) = \sum_{j=0}^{N_{\mathrm{a}}} \sum_{k=\lfloor \frac{N_{\mathrm{a}}}{2} \rfloor + 1}^{N_{\mathrm{a}}} P(N_{\mathrm{cor}}^{(2)} = k | N_{\mathrm{cor}}^{(1)} = j) P(N_{\mathrm{cor}}^{(1)} = j)$$

$$= \sum_{j=0}^{N_{\mathrm{a}}} P(N_{\mathrm{cor}}^{(1)} = j) \sum_{k=\lfloor \frac{N_{\mathrm{a}}}{2} \rfloor + 1}^{N_{\mathrm{a}}} \binom{N_{\mathrm{a}}}{k} e^{\alpha k(j-N_{\mathrm{a}})} (1 - e^{\alpha(j-N_{\mathrm{a}})})^{N_{\mathrm{a}}-k},$$

where $P(N_{\mathrm{cor}}^{(1)} = j) = \binom{N_{\mathrm{a}}}{j} p^j (1-p)^{N_{\mathrm{a}}-j}$.

Then, similar to Proposition 2.2, if $e^{\alpha(j-N_{\mathrm{a}})} \geq \frac{1}{2}$ (i.e., $j \geq N_{\mathrm{a}} - \frac{1}{\alpha} \ln 2$), we have the lower bound:

$$P(N_{\mathrm{cor}}^{(2)} > \frac{N_{\mathrm{a}}}{2}) \geq \sum_{j=\lceil N_{\mathrm{a}} - \frac{\ln 2}{\alpha} \rceil}^{N_{\mathrm{a}}} \binom{N_{\mathrm{a}}}{j} p^j (1-p)^{N_{\mathrm{a}}-j} \left( 1 - \exp\left( -2N_{\mathrm{a}} \left( \frac{1}{2} - e^{\alpha(j-N_{\mathrm{a}})} \right)^2 \right) \right).$$

Otherwise, if $e^{\alpha(j-N_{\mathrm{a}})} < \frac{1}{2}$ (i.e., $j < N_{\mathrm{a}} - \frac{1}{\alpha} \ln 2$), we have:

$$P(N_{\mathrm{cor}}^{(2)} > \frac{N_{\mathrm{a}}}{2}) = \sum_{j=0}^{\lfloor N_{\mathrm{a}} - \frac{\ln 2}{\alpha} \rfloor} \binom{N_{\mathrm{a}}}{j} p^j (1-p)^{N_{\mathrm{a}}-j} \sum_{k=\lfloor \frac{N_{\mathrm{a}}}{2}+1 \rfloor}^{N_{\mathrm{a}}} \binom{N_{\mathrm{a}}}{k} e^{\alpha k(j-N_{\mathrm{a}})} (1 - e^{\alpha(j-N_{\mathrm{a}})})^{N_{\mathrm{a}}-k}$$

$$+ \sum_{j=\lceil N_{\mathrm{a}} - \frac{\ln 2}{\alpha} \rceil}^{N_{\mathrm{a}}} \binom{N_{\mathrm{a}}}{j} p^j (1-p)^{N_{\mathrm{a}}-j} \sum_{k=\lfloor \frac{N_{\mathrm{a}}}{2}+1 \rfloor}^{N_{\mathrm{a}}} \binom{N_{\mathrm{a}}}{k} e^{\alpha k(j-N_{\mathrm{a}})} (1 - e^{\alpha(j-N_{\mathrm{a}})})^{N_{\mathrm{a}}-k}.$$

Since the case $j \geq N_{\mathrm{a}} - \frac{1}{\alpha} \ln 2$ is not considered here, we simply bound the probability $P(N_{\mathrm{cor}}^{(2)} > \frac{N_{\mathrm{a}}}{2} \mid N_{\mathrm{cor}}^{(1)} = j)$ by 1. Thus, we then have:

$$P(N_{\mathrm{cor}}^{(2)} > \frac{N_{\mathrm{a}}}{2}) \leq \sum_{j=0}^{\lfloor N_{\mathrm{a}} - \frac{\ln 2}{\alpha} \rfloor} \binom{N_{\mathrm{a}}}{j} p^j (1-p)^{N_{\mathrm{a}}-j} \sum_{k=\lceil \frac{N_{\mathrm{a}}}{2} \rceil + 1}^{N_{\mathrm{a}}} \binom{N_{\mathrm{a}}}{k} e^{\alpha k(j-N_{\mathrm{a}})} (1 - e^{\alpha(j-N_{\mathrm{a}})})^{N_{\mathrm{a}}-k}$$

$$+ \sum_{j=\lceil N_{\mathrm{a}} - \frac{\ln 2}{\alpha} \rceil}^{N_{\mathrm{a}}} \binom{N_{\mathrm{a}}}{j} p^j (1-p)^{N_{\mathrm{a}}-j}.$$

$$P(N_{\mathrm{cor}}^{(2)} > \frac{N_{\mathrm{a}}}{2}) \leq \sum_{j=0}^{\lfloor N_{\mathrm{a}} - \frac{\ln 2}{\alpha} \rfloor} \binom{N_{\mathrm{a}}}{j} p^j (1-p)^{N_{\mathrm{a}}-j} \exp\left( -2N_{\mathrm{a}} \left( \frac{1}{2} - e^{\alpha(j-N_{\mathrm{a}})} \right)^2 \right)$$

$$+ \sum_{j=\lceil N_{\mathrm{a}} - \frac{\ln 2}{\alpha} \rceil}^{N_{\mathrm{a}}} \binom{N_{\mathrm{a}}}{j} p^j (1-p)^{N_{\mathrm{a}}-j}.$$

The proof is completed. $\qquad\square$

## C    DETAILED ANALYSIS ON TOKEN CONSUMPTION

Denote the token of the query $x^{\text{test}}$ as $T^{\text{q}}$, the token of the output of the agent $A_{\theta_i}$ at the $r$-th debate round as $T^{\text{o}}_{i,r}$. Then, we formulate the token consumption of both MAD and MAD-M$^2$ as follows.

For MAD, at the initial round, the query $x^{\text{test}}$ is fed into $N_{\text{a}}$ LLM agents and then $N_{\text{a}}$ responses are generated from these LLM agents. Thus, the consumption of MAD at the initial debate round is:

$$N_1^{\text{token}} = N_{\text{a}} T^{\text{q}} + \sum_{i=1}^{N_{\text{a}}} T^{\text{o}}_{1,i}.$$

Then, from the second debate round, each agent will take all $N_{\text{a}}$ outputs in the last debate round and output a new reasoning response with the query. Thus, the consumption of tokens for MAD at the $r$-th debate round is:

$$N_r^{\text{token}} = N_{\text{a}} \left( T^{\text{q}} + \sum_{i=1}^{N_{\text{a}}} T^{\text{o}}_{r-1,i} \right) + \sum_{i=1}^{N_{\text{a}}} T^{\text{o}}_{r,i}.$$

Thus, the total consumption of tokens of MAD can be formulated as:

$$N_{\text{MAD}}^{\text{token}} = \sum_{r=1}^{N_{\text{round}}} N_r^{\text{token}}$$

$$= N_{\text{a}} N_{\text{round}} T^{\text{q}} + N_{\text{a}} \sum_{r=2}^{N_{\text{round}}} \sum_{i=1}^{N_{\text{a}}} T^{\text{o}}_{r-1,i} + \sum_{r=1}^{N_{\text{round}}} \sum_{i=1}^{N_{\text{a}}} T^{\text{o}}_{r,i}.$$

For MAD-M$^2$, at the initial round, the consumption of tokens is the same as MAD:

$$N_1^{\text{token}} = N_{\text{a}} T^{\text{q}} + \sum_{i=1}^{N_{\text{a}}} T^{\text{o}}_{1,i}.$$

From the second round, the agents will first evaluate the previous memories and mask the potentially incorrect memories. In this step, all memories are fed into each agent, and the agent will output "yes", "no", or "unsure". Here, we ignore the tokens of outputs and the instructions. We then formulate the consumption of tokens at this step as:

$$N_r^{\text{eval\_token}} = N_{\text{a}} \sum_{i=1}^{N_{\text{a}}} T^{\text{o}}_{r-1,i}.$$

Then, based on the selected memories $\widehat{\mathcal{M}}^{(i)}_{r-1}$, the input tokens for each agent in the $r$-th debate round should be no more than the sum of the query tokens and the tokens of all previous tokens. Thus, considering the output tokens, we have:

$$N_r^{\text{token}} \leq N_{\text{a}} \left( T^{\text{q}} + \sum_{i=1}^{N_{\text{a}}} T^{\text{o}}_{r-1,i} \right) + \sum_{i=1}^{N_{\text{a}}} T^{\text{o}}_{r,i}.$$

Thus, the total consumption of MAD-M$^2$ can be formulated as:

$$N_{\text{MAD}-\text{M}^2}^{\text{token}} = N_1^{\text{token}} + \sum_{r=2}^{N_{\text{round}}} \left( N_r^{\text{eval\_token}} + N_r^{\text{token}} \right)$$

$$\leq N_{\text{a}} T^{\text{q}} + \sum_{i=1}^{N_{\text{a}}} T^{\text{o}}_{1,i} + \sum_{r=2}^{N_{\text{round}}} \left( N_{\text{a}} \sum_{i=1}^{N_{\text{a}}} T^{\text{o}}_{r-1,i} + N_{\text{a}} \left( T^{\text{q}} + \sum_{i=1}^{N_{\text{a}}} T^{\text{o}}_{r-1,i} \right) + \sum_{i=1}^{N_{\text{a}}} T^{\text{o}}_{r,i} \right)$$

$$= N_{\text{a}} N_{\text{round}} T^{\text{q}} + 2N_{\text{a}} \sum_{r=2}^{N_{\text{round}}} \sum_{i=1}^{N_{\text{a}}} T^{\text{o}}_{r-1,i} + \sum_{r=1}^{N_{\text{round}}} \sum_{i=1}^{N_{\text{a}}} T^{\text{o}}_{r,i}.$$

## D COMPLETE EXPERIMENTAL SETTINGS

In this section, we provide more detailed experimental settings adopted in this paper to ensure that the results reported in this paper are reproducible. The content here refers to the details of adopted LLMs, evaluation benchmarks, hyperparameters, hardware, and reasoning/debate/masking prompts.

### D.1 DETAILS OF PROMPTS

In this section, we provide comprehensive details regarding the prompts adopted in the experiments of our proposed MAD-$M^2$ framework. Specifically, we provide the prompts adopted in the CoT reasoning, the multi-agent debate process, and the subjective evaluation operation on memories.

---

**CoT Prompt**

`<QUESTION>`
Please solve the problem step by step.

### Response format (MUST be strictly followed) (DO NOT include any other formats except for the given XML format):
<think>YOUR THINKING HERE</think>
<answer>YOUR FINAL ANSWER ONLY, NO OTHER TEXT</answer>.

---

**Debate Prompt**

These are the potential solutions to the problem:
`<CONTEXT>`
Use the potential solutions as additional information for the following question.
Question:`<QUESTION>`
Please think step by step and solve the problem.
### Response format (MUST be strictly followed) (DO NOT include any other formats except for the given XML format):
<think>YOUR THINKING HERE</think>
<answer>YOUR FINAL ANSWER ONLY, NO OTHER TEXT</answer>.

---

**Self-Evaluation Prompt**

Evaluate the given solutions based on the question. ** Your response MUST end with the following format: <label>YES</label> or <label>NO</label> or <label>NOT SURE</label>. ** Return YES if the solution is completely correct, NO if any part of the solution is incorrect, and NOT SURE if you are unsure.
Question: `<QUESTION>`
Solutions: `<SOLUTION>`

---

### D.2 BASELINES

To validate the effectiveness of our proposed MAD-$M^2$, the following reasoning frameworks are adopted as baselines in this paper: (1) Chain-of-Thought (CoT) (Wei et al., 2022b); (2) Self-Consistency Chain-of-Thoughts (CoT-SC) (Wang et al., 2023) with 6 independent reasoning paths; and (3) Multi-Agent Debate (MAD, 3-agent 2-round) (Du et al., 2023). As a default setting, an MAD framework is composed of 3 LLM agents and 2 debate rounds if there are no specific clarifications.

### D.3 LARGE LANGUAGE MODELS

In this work, to validate the performance of our proposed MAD-$M^2$, we mainly consider evaluating baseline reasoning/multi-agent debate frameworks and our proposed MAD-$M^2$ with open-source large language models with different sizes. Specifically, we adopt Qwen2.5-7B-Instruct, Qwen2.5-Math-7B-Instruct (Yang et al., 2024a;b), DeepSeek-Math-7B-Instruct (Shao et al., 2024), and QwQ-32B (Team, 2025) in all our experiments and analyses.

### D.4 GENERAL TEST DATA SETTINGS

In all experiments of this paper, we evaluate all reasoning frameworks with trials under five different random seeds (i.e., 41-45) and report the average of all five runs as the final results. In each run, for GSM8K, MATH, and MMLU_Pro, we randomly sample 100 samples from a shuffled sequence of the original benchmark due to the computational budgets. Such a setting is consistent with that in the previous work (Du et al., 2023). In contrast, for AIME 24 & 25, we adopt all questions for evaluation.

### D.5 HYPERPARAMETER SETTINGS

**Hyperparameters of LLMs.** In this paper, all reasoning responses are performed on LLMs with the temperature set to 1.0 except for Qwen2.5-Math-7B-Instruct, since there are many unrecognized characters when performing reasoning with Qwen2.5-Math-7B-Instruct with the temperature set to 1.0. Thus, to avoid failure in reasoning, the temperature for Qwen2.5-Math-7B-Instruct is set to 0.2. Moreover, the `top_p` is set to 1.0 for all cases. The maximum length of the token is set to 24064 for Qwen2.5-7B-Instruct, 4096 for Qwen2.5-Math-7B-Instruct and DeepSeek-Math-7B-Instruct, and 131072 for QwQ-32B. Since the length of output tokens of QwQ-32B is usually too long, the thinking tokens in outputs will be truncated. Specifically, only the first 4096 tokens are preserved as memories.

**Fairness of Comparison between CoT-SC and MAD.** In the experiments of our paper, the default setting for MAD includes 3 agents and 2 debate rounds. Thus, the LLMs in MAD will consume the resources of 6 reasoning trajectory generations. Thus, for fairness, in the case of CoT-SC, the final answer is achieved by performing majority voting among 6 responses generated in the way of CoT.

**Computational Resources.** In this paper, the main experiments and the analyses regarding Qwen2.5-7B-Instruct, Qwen2.5-Math-7B-Instruct, and DeepSeek-Math-7B-Instruct are conducted with two RTX 3090 GPUs, while the experiments of QwQ-32B are conducted on several A100 and H20 GPUs.

### D.6 DETAILS OF EVALUATION BENCHMARKS

In this paper, we adopt 5 mainstream benchmarks, ranging from mathematical reasoning to language understanding tasks, to evaluate all baselines and our proposed MAD-$M^2$. The mathematical reasoning benchmarks include GSM8K, MATH, AIME 24, and AIME 25, while the language understanding benchmark is MMLU_Pro. Details about these benchmarks are provided as follows.

- **AIME 24 & 25:** AIME 24 & 25 are challenging competition mathematical question sets on the 2024 and 2025 American Invitational Mathematics Examinations. Each dataset contains 30 hard mathematical problems. These two benchmarks are famous for their difficulty in reasoning.
- **GSM8K (Cobbe et al., 2021):** GSM8K (a.k.a. Grade School Math 8K) is a popular mathematical reasoning dataset consisting of about 8500 high-quality, diverse grade school math word problems. In our experiments, we only adopt its test set that is composed of 1,318 questions for evaluation.
- **MATH (Hendrycks et al., 2021):** MATH is a comprehensive mathematical dataset designed to rigorously challenge the mathematical reasoning abilities of LLMs. The questions in the MATH dataset refer to algebra, geometry, number theory, and combinatorics, thereby thoroughly evaluating the mathematical understanding and problem-solving capabilities of LLMs. In the experiments of this paper, we only adopt its 5,000 test data to evaluate the baselines and our proposed MAD-$M^2$.
- **MMLU_Pro (Wang et al., 2024):** MMLU-Pro is an advanced extension of the Massive Multitask Language Understanding (MMLU) dataset (Hendrycks et al., 2020), encompassing 57 diverse tasks across domains such as mathematics, computer science, biology, and physics. Compared to MMLU, MMLU-Pro introduces more complex, reasoning-intensive problems, which are more challenging.

## E COMPLETE EXPERIMENTAL RESULTS

In this section, we provide more supplementary experimental results of those in Section 4 and more analysis results regarding our proposed MAD-$M^2$ to validate its efficacy and explore its properties.

### E.1 COMPLETE ANALYSES ON SCALING THE NUMBER OF AGENTS AND DEBATE ROUNDS

In Section 4.3, we have conducted experiments to figure out the scaling capability of our proposed MAD-$M^2$equipped with different LLMs (i.e., Qwen2.5-7B-Instruct and DeepSeek-Math-7B-Instruct). Generally, according to the results in Section 4.3, MAD-$M^2$ consistently benefits from the scaling of the number of agents, while the performance fluctuates as the number of debate rounds increases. Moreover, another important observation here is that the subjective masking strategy performs

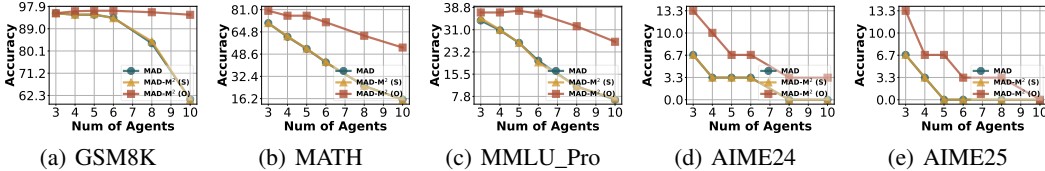

(a) GSM8K      (b) MATH      (c) MMLU_Pro      (d) AIME24      (e) AIME25

Figure 8: **Effect of scaling the number of agents in the case of Qwen2.5-Math-7B-Instruct.** The number of agents is increased from 3 to 10. According to the figures, both frameworks act differently when the number of agents increases and MAD-M$^2$(O) achieves better performance in most cases.

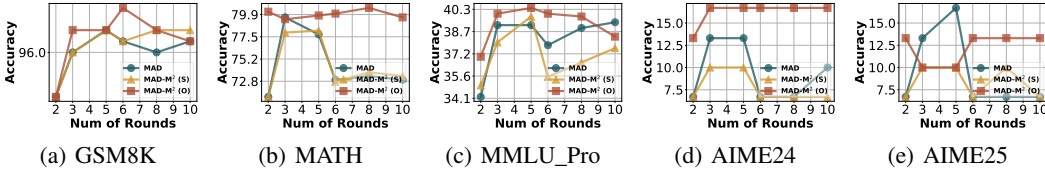

(a) GSM8K      (b) MATH      (c) MMLU_Pro      (d) AIME24      (e) AIME25

Figure 9: **Effect of scaling the number of debate rounds w.r.t. Qwen2.5-Math-7B-Instruct.** The number of debate rounds increases from 2 to 10. According to the figures, the performance of different cases varies. In most cases, MAD-M$^2$(O) achieves better performance than MAD-M$^2$(S).

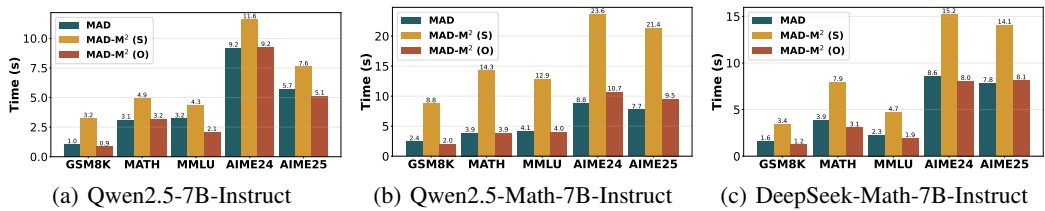

(a) Qwen2.5-7B-Instruct      (b) Qwen2.5-Math-7B-Instruct      (c) DeepSeek-Math-7B-Instruct

Figure 10: **Analyses on time consumption of MAD and MAD-M$^2$ with different LLMs.** Each figure compares the time consumption of MAD, MAD-M$^2$(S), and MAD-M$^2$(O). Generally, MAD-M$^2$(O) consumes less time than MAD-M$^2$(S) since it is free of an additional self-evaluation step.

better on MAD-M$^2$ equipped with weak LLMs while the objective masking strategy achieves better performance on MAD-M$^2$ equipped with powerful models. Here, we further visualize the analyses of MAD-M$^2$ with different masking strategies on Qwen2.5-Math-7B-Instruct (cf. Figs. 8 and 9).

According to the visualization results, we can observe that the objective masking strategy consistently achieves better performance than the subjective masking strategy when scaling the number of agents and debate rounds. This phenomenon aligns with that in Section 4.3. However, different from the case of MAD-M$^2$ with Qwen2.5-7B-Instruct and DeepSeek-Math-7B-Instruct, the performance of MAD-M$^2$ drops when scaling the number of agents. Compared to MAD-M$^2$(S), MAD-M$^2$(O) is more robust in the performance degradation. According to our observation on the behavior of the LLM, the reason for this phenomenon is the failure of reasoning due to the overlong output sequence.

### E.2 TIME CONSUMPTION ANALYSIS

Compared to the conventional MAD framework, MAD-M$^2$ introduces a self-evaluation and masking phase to remove the potential erroneous memories and further improve the performance of multi-agent debate. However, a concern raised here is whether the efficiency of MAD is negatively influenced.

To figure out this concern, we compared the time consumption of MAD, MAD-M$^2$(S), and MAD-M$^2$(O) on the five benchmarks with different LLMs. The quantitative results are visualized in Fig. 10. According to the results in figures, we can observe that MAD-M$^2$(O) consumes comparable time to the baseline, MAD, while much less time than MAD-M$^2$(S). Such a gap probably results from the additional self-evaluation operation in the subjective masking strategy. Meanwhile, compared to those simple reasoning tasks (e.g., GSM8K, MATH, and MMLU_Pro), more time is consumed on harder reasoning tasks (e.g., AIME). Thus, for the case of MAD-M$^2$ with powerful LLMs, our proposed MAD-M$^2$(with the objective masking strategy) is effective and efficient in reasoning tasks.

### E.3 INVESTIGATION ON STRICTNESS IN SUBJECTIVE MASKING STRATEGY

In our proposed MAD-M$^2$, an important component is the evaluation and masking. In particular, in the subjective masking strategy, the memories are evaluated by agents between two rounds of

Table 2: **Analysis on strictness of MAD-M$^2$(S).** Average accuracy with standard deviation is reported. We evaluate three small LLMs with both mathematical reasoning and language understanding benchmarks. For fairness, all results are the average of five trials on different seeds (i.e., 41-45).

| Method | AIME24 | AIME25 | MMLU_Pro | MATH | GSM8K |
|---|---|---|---|---|---|
| Qwen2.5-7B-Instruct | | | | | |
| MAD-M$^2$(S) | 13.3 | 3.3 | 43.6±2.3 | 56.8±2.1 | 89.0±4.0 |
| MAD-M$^2$(S)-Strict | 10.0 (-3.3) | 10.0 (+6.7) | 43.4±5.3 (-0.2) | 54.8±2.8 (-2.0) | 87.6±3.4 (-1.4) |
| Qwen2.5-Math-7B-Instruct | | | | | |
| MAD-M$^2$(S) | 6.7 | 6.7 | 35.0±2.2 | 71.2±3.3 | 95.2±1.8 |
| MAD-M$^2$(S)-Strict | 13.3 (+6.6) | 6.7 (+0.0) | 40.8±4.4 (+5.8) | 81.4±4.4 (+10.2) | 95.4±1.5 (+0.2) |
| DeepSeek-Math-7B-Instruct | | | | | |
| MAD-M$^2$(S) | 0.0 | 0.0 | 30.8±5.2 | 37.0±5.1 | 80.8±3.5 |
| MAD-M$^2$(S)-Strict | 0.0 (+0.0) | 0.0 (+0.0) | 26.2±4.7 (-4.6) | 40.0±5.1 (+3.0) | 82.6±2.9 (+1.8) |

Table 3: **Comparison between MAD-M$^2$ and Sparse MAD.** Average accuracy with standard deviation is reported. We compare MAD-M$^2$(S), MAD-M$^2$(O), and Sparse MAD on Qwen2.5-7B-Instruct and Qwen2.5-Math-7B-Instruct with both mathematical reasoning and language understanding benchmarks. For fairness, all results are the average of five trials on different seeds (i.e., 41-45).

| Method | AIME24 | AIME25 | MMLU_Pro | MATH | GSM8K |
|---|---|---|---|---|---|
| Qwen2.5-7B-Instruct | | | | | |
| MAD-M$^2$(S) | 16.7 | **13.3** | **44.6±5.7** | **60.8±1.9** | 93.6±1.1 |
| MAD-M$^2$(O) | **20.0** | 0.0 | 42.2±4.2 | 58.8±5.6 | 92.8±1.3 |
| Sparse MAD | 13.3 | 6.7 | 44.2±3.4 | 59.2±2.7 | **94.2±1.9** |
| Qwen2.5-Math-7B-Instruct | | | | | |
| MAD-M$^2$(S) | **16.7** | 13.3 | 66.2±4.6 | 62.8±4.3 | **96.4±1.8** |
| MAD-M$^2$(O) | 13.3 | **16.7** | **67.4±7.0** | 63.8±3.0 | 95.8±2.7 |
| Sparse MAD | 10.0 | 10.0 | 38.6±2.3 | **76.4±3.8** | 96.2±1.8 |

debates. In our MAD-M$^2$ framework, when applying the subjective masking strategy, the memories are evaluated with flags "YES", "NO", and "NOT SURE". Typically, the "YES" and "NO" flags are mapped into True and False, respectively. For the "NOT SURE" flag, according to the strictness, it is mapped into True or False. Thus, to figure out the effect of the strictness, we compare the performance of MAD-M$^2$(S) with and without the strictness. The results are reported in Table 2.

According to the results, we can observe that the performance of MAD-M$^2$(S) varies among LLMs. Specifically, in the case of MAD-M$^2$(S) with weak LLMs (e.g., Qwen2.5-7B-Instruct), applying strict rules to evaluation deteriorates the performance. For example, the performance on MATH and GSM8K drops by 2.0% and 1.4%, respectively. However, in the case of MAD-M$^2$(S) with powerful LLMs (e.g., Qwen2.5-Math-7B-Instruct), the performance increases by 5.8%, 10.2%, and 0.2%, respectively, on MMLU_Pro, MATH, and GSM8K. In summary, MAD-M$^2$(S) with powerful LLMs benefits from the strict rules, while MAD-M$^2$(S) fails to achieve better performance with strict rules.

