# OpenReview forum: "Multi-Agent Debate with Memory Masking"
_ICLR.cc/2026/Conference — ICLR 2026 Poster_

### Official Review · Reviewer_gCX4 · 2025-10-29

**Soundness:** 3
**Presentation:** 3
**Contribution:** 2
**Rating:** 4
**Confidence:** 4

**Summary:**

This paper investigates a fundamental limitation of the Multi-Agent Debate (MAD) framework, demonstrating that LLM agents exhibit vulnerability to erroneous memories from preceding debate rounds, which can propagate incorrect reasoning trajectories. Through rigorous theoretical analysis, the authors establish that MAD performance is intrinsically bounded by e^(-αNₑ), where Nₑ denotes the cardinality of erroneous memories and α characterizes the agent's robustness coefficient. To address this vulnerability, they propose MAD-M² (Multi-Agent Debate with Memory Masking), a framework that incorporates a memory evaluation and masking mechanism, enabling agents to selectively filter low-quality contextual information prior to each reasoning iteration.

**Strengths:**

1. This paper is the first to demonstrate that Multi-Agent Debate (MAD) frameworks are vulnerable to false memories, providing theoretical proof of this phenomenon.
2. This study offers a thorough analysis of the key characteristics of the proposed MAD-M² framework in relation to existing methods.

**Weaknesses:**

1. The primary contribution of this paper lies in its theoretical insights rather than methodological innovations.

2. The proposed MAD-M² framework achieves state-of-the-art performance in only a limited subset of experimental scenarios, while underperforming traditional MAD or CoT-SC methods in the majority of cases. Moreover, MAD-M² incurs significantly higher token consumption compared to baseline methods. From a performance-efficiency trade-off perspective, MAD-M² does not demonstrate clear contributions. If additional metrics can substantiate the contributions of MAD-M², please include them in Table 1.

**Questions:**

Refer to the weaknesses.

---

> ### Author Response · Authors · 2025-11-23
>
> - **Performance on Qwen-2.5-7B-Instruct**
> In this experiment, we evaluate Qwen-2.5-7B on Math, MMLU_Pro, AIME24, and AIME25 datasets. Each case is the average of 4 runs with different seeds. Among all methods, **MAD** denotes the naive multi-agent debate method, **MAD-Naive** denotes the multi-agent debate with **subjectively pruning strategy**, and **MAD-PPL** denotes the multi-agent debate with **objectively pruning strategy (perplexity)**.
> | **Methods** | **MATH** | **MMLU_Pro** | **AIME24** | **AIME25** |
> | ------ | ------ | ------ | ------ | ------ |
> | **CoT** | 0.470$\pm$0.05 | 0.403$\pm$0.07 | 0.067 | 0.033 |
> | **CoT-SC** | **0.548$\pm$0.03** | **0.420$\pm$0.04** | 0.1 | **0.067** |
> | **MAD** | 0.485$\pm$0.03 | 0.395$\pm$0.03 | 0.1 | 0.033 |
> | **MAD-Naive** | *0.510$\pm$0.01* | *0.410$\pm$0.03* | **0.13** | **0.067**  |
> | **MAD-PPL** | 0.488$\pm$0.03 | 0.395$\pm$0.03 | 0.1 | 0.033 |
>
> - **Token Consumption**
> | **Methods** | **MATH** | **MMLU_Pro** | **AIME24** | **AIME25** |
> | ------ | ------ | ------ | ------ | ------ |
> | **CoT** | 57613 | 59034 | 30387 | 29907|
> | **CoT-SC** | 352900 | 360260 | 181639 | 173850 |
> | **MAD** | 347143 | 360029 | 179748 | 178399 |
> | **MAD-Naive** | 567655 | 567589 | 285043 | 279145  |
> | **MAD-PPL** | 347143 | 360029 | 179748 | 178399 |

---

> ### Author Response · Authors · 2025-11-23
>
> Dear Reviewer gCX4,
>
> Thank you for your time and efforts in reviewing our paper. We really appreciate your constructive and insightful comments and suggestions. According to your concerns and questions, we provide responses as follows.
>
> > W1. The primary contribution of this paper lies in its theoretical insights rather than methodological innovations.
>
> **Answer:** Thank you for your recognition of our theoretical insights. However, we think that our contributions are not merely limited to theory. To address your concern about our contributions in this paper, we provide detailed explanations from the perspectives of the problem, the theory, and the methodology.
>
> - **_[Problem Perspective.]_** The first contribution of our paper is that **we observe that LLM agents in the conventional MAD framework are vulnerable to erroneous responses in the previous round**. This indicates that the conventional MAD framework is not robust enough. Since robustness is an important aspect of building agent frameworks, the problem studied in this paper is non-trivial.
> - **_[Theory Perspective.]_** The second contribution of our paper is that **we build theoretical analyses on both CoT-SC and MAD reasoning frameworks to explore their performance boundaries**. In our theoretical analyses, we examine the performance boundaries of CoT-SC and MAD in detail under different reasoning problem settings. In all cases, we find that enhancing the capability of LLM agents is an essential way to improve the performance of the reasoning framework. With the assumption that the capability of LLM agents is affected by the memories in the previous debate round, the analyses motivate us to improve the MAD framework via masking the erroneous reasoning trajectories in the previous rounds. From this perspective, our proposed MAD-M$^2$ is well supported and motivated by theoretical results.
> - **_[Methodology Perspective.]_** Inspired by Proposition 2.3 in our paper, we found that a feasible way to improve the performance of MAD in both hard and easy reasoning problem settings is improving the capability of LLM agents in the frameworks. Specifically, we need to enhance the probability that LLMs answer the questions correctly in the second round (i.e., $e^{-\alpha N_{\rm e}}$). Since the probability is related to the number of erroneous responses in the previous round, a simple and direct way is to filter out those incorrect responses.
>
> Overall, we would like to clarify here that **MAD-M$^2$ is not merely a simple variant of MAD**. **Actually, MAD-M$^2$ introduces a new mechanism in the debate round**. Different from conventional MAD, which blindly feeds all memories in the previous round into the next round, in MAD-M$^2$, each agent is required to **(1) explicitly evaluate all memories**; **(2) generate round-level agent-specific binary masks that determine which memories are adopted as the next-round context**. **Such a conceptually simple and lightweight "evaluation-masking-reasoning" cycle is a universal component that can be plugged into any MAD system, independent of models, number of agents, or topology**. From this perspective, the simplicity of MAD-M$^2$ can be viewed as a strength rather than a limitation.

---

> ### Author Response · Authors · 2025-11-23
>
> > W2. The proposed MAD-M² framework achieves state-of-the-art performance in only a limited subset of experimental scenarios, while underperforming traditional MAD or CoT-SC methods in the majority of cases. Moreover, MAD-M² incurs significantly higher token consumption compared to baseline methods. From a performance-efficiency trade-off perspective, MAD-M² does not demonstrate clear contributions. If additional metrics can substantiate the contributions of MAD-M², please include them in Table 1.
>
> **Answer:** Thank you for your comments. According to your comments, we summarize your concerns as follows: **(1) MAD-M$^2$ fails to outperform baselines, such as MAD and CoT-SC in most cases**; and **(2) MAD-M$^2$ consumes more tokens than baselines**.
>
> First of all, **we would clarify that our proposed MAD-M$^2$ outperforms MAD baseline in most cases**. In the DeepSeek-V3 case, MAD-M$^2$ outperforms MAD on both AIME24 (+0.13) and MATH (+0.02) benchmarks while achieving comparable performance on AIME25 and MMLU_Pro. In the Qwen2.5-72B case, MAD-M$^2$ outperforms MAD on both MMLU_Pro (+0.03) and MATH (+0.02) benchmarks while underperforming MAD on AIME benchmarks. In the Mixtral-8x7B case, MAD-M$^2$ outperforms MAD on both MMLU_Pro (+0.03) and MATH (+0.04) benchmarks while achieving comparable results on AIME benchmarks.
>
> To further address your first concern, we propose to evaluate our proposed MAD-M$^2$ (MAD-M$^2$ (Naive)) and all baselines on all benchmarks with Qwen-2.5-7B-Instruct. Here, we also explore an extra pruning strategy, perplexity, in our proposed MAD-M$^2$ method. The performance and token consumptions are reported in the tables above. According to the results reported in the table, we can observe that MAD-M$^2$ (Naive) outperforms MAD in all cases. Specifically, **MAD-M$^2$ (Naive) achieves 2.5%, 1.5%, 3%, and 3.4% improvements on MATH, MMLU_Pro, AIME24, and AIME25, respectively**. Moreover, **on AIME24 and AIME25, MAD-M$^2$ achieves comparable or even better performance than CoT-SC**. All these experiments demonstrate the effectiveness of our proposed method. Meanwhile, we also notice that MAD-M$^2$ (PPL) achieves comparable results to MAD, which indicates that perplexity is not an appropriate tool for memory masking since it implicitly assumes that the correctness and the perplexity are linearly relevant.
>
> To address your second concern, we would like to clarify that **one of the reasons for the higher token consumption is that we add `try...except..` sections in our codes**. To further figure out the token consumption in our proposed MAD-M$^2$ method, we remove the `try...except...` parts in our experiments on Qwen-2.5-7B-Instruct. According to our empirical results (see the second table above), the token consumption of MAD-M$^2$ (Naive) is more than MAD. Specifically, **MAD-M$^2$ consumes 63.5%, 57.7%, 58.6%, and 56.5% more tokens than MAD. Such a phenomenon is consistent with our theoretical analysis in our paper (see token consumption analysis in Section 3.2)**. According to our results, we find that such consumption results from the long responses of the LLMs. We will provide detailed discussions about this phenomenon in our updated version.

---

### Official Review · Reviewer_7GkP · 2025-11-01

**Soundness:** 2
**Presentation:** 3
**Contribution:** 2
**Rating:** 4
**Confidence:** 3

**Summary:**

This paper investigates the robustness issue in the multi-agent debate (MAD) framework for large language model (LLM) reasoning. The authors theoretically demonstrate that the performance of MAD degrades when agents rely on “wrong memories” from prior debate rounds, since each round’s reasoning depends on previous outputs. To mitigate this, they propose MAD-M2 (Multi-Agent Debate with Memory Masking), which allows each agent to evaluate and selectively mask unreliable memories before generating new responses. The method is conceptually simple yet well-motivated. Experiments across four reasoning benchmarks and diverse models show consistent or improved results compared with vanilla MAD.

**Strengths:**

1. Clear motivation grounded in a theoretical gap: The authors identify an underexplored limitation in multi-agent debate frameworks — vulnerability to low-quality reasoning memories — and back it up with intuitive examples (Figure 1) and formal analysis.

2. The analysis in Section 2 provides explicit probabilistic bounds linking performance degradation to the number of wrong memories. The inclusion of both hard and easy reasoning settings (HPR/EPR) offers a nuanced interpretation of how memory errors affect debate robustness.

**Weaknesses:**

1. My biggest concerns are about approach performance. I only see the marginal performance gains in stronger models. While MAD-M2 improves certain benchmarks on easy tasks, the improvements for DeepSeek-V3 are minor or inconsistent. This suggests limited scalability to highly capable LLMs. Could the authors provide more results on diverse and powerful LLM benchmarks?

2. The proposed masking step roughly doubles token consumption in some cases, but the paper provides limited quantitative analysis of efficiency–performance trade-offs.

3. The masking mechanism is described at a high level but not deeply analyzed. For instance, how different scoring heuristics or thresholds affect the results remains unclear. The authors are suggested to add more fine-grained ablations (e.g., varying mask strictness).

4. Although the paper compares conceptually to S-MAD and S2-MAD, it doesn’t empirically benchmark against them.

**Questions:**

1. How sensitive is MAD-M2 to the rule for generating the binary mask?

2. Did the authors explore partial masking or soft weighting? Could such approaches improve stability?

3. Can the authors provide more comprehensive experiments on more LLM backbones and compare more baseline MAD approaches?

---

> ### Author Response · Authors · 2025-11-23
>
> - **Performance on Qwen-2.5-7B-Instruct**
> In this experiment, we evaluate Qwen-2.5-7B on Math, MMLU_Pro, AIME24, and AIME25 datasets. Each case is the average of 4 runs with different seeds. Among all methods, **MAD** denotes the naive multi-agent debate method, **MAD-M$^2$ (Naive)** denotes the multi-agent debate with **subjectively pruning strategy**, and **MAD-M$^2$ (PPL)** denotes the multi-agent debate with **objectively pruning strategy (perplexity)**.
> | **Methods** | **MATH** | **MMLU_Pro** | **AIME24** | **AIME25** |
> | ------ | ------ | ------ | ------ | ------ |
> | **CoT** | 0.470$\pm$0.05 | 0.403$\pm$0.07 | 0.067 | 0.033 |
> | **CoT-SC** | **0.548$\pm$0.03** | **0.420$\pm$0.04** | 0.1 | **0.067** |
> | **MAD** | 0.485$\pm$0.03 | 0.395$\pm$0.03 | 0.1 | 0.033 |
> | **MAD-M$^2$ (Naive)** | *0.510$\pm$0.01* | *0.410$\pm$0.03* | **0.13** | **0.067**  |
> | **MAD-M$^2$ (PPL)** | 0.488$\pm$0.03 | 0.395$\pm$0.03 | 0.1 | 0.033 |
>
> - **Token Consumption**
> | **Methods** | **MATH** | **MMLU_Pro** | **AIME24** | **AIME25** |
> | ------ | ------ | ------ | ------ | ------ |
> | **CoT** | 57613 | 59034 | 30387 | 29907|
> | **CoT-SC** | 352900 | 360260 | 181639 | 173850 |
> | **MAD** | 347143 | 360029 | 179748 | 178399 |
> | **MAD-M$^2$ (Naive)** | 567655 | 567589 | 285043 | 279145  |
> | **MAD-M$^2$ (PPL)** | 347143 | 360029 | 179748 | 178399 |

---

> ### Author Response · Authors · 2025-11-23
>
> Dear Reviewer 7GkP,
>
> Thank you for your time and efforts in reviewing our paper. We really appreciate your constructive and insightful comments and suggestions. According to your concerns and questions, we provide responses as follows.
>
> > W1. My biggest concerns are about approach performance. I only see the marginal performance gains in stronger models. While MAD-M2 improves certain benchmarks on easy tasks, the improvements for DeepSeek-V3 are minor or inconsistent. This suggests limited scalability to highly capable LLMs. Could the authors provide more results on diverse and powerful LLM benchmarks?
>
> **Answer:** Thank you for your comments. Based on your comments, we summarize your main concern here is that our proposed MAD-M$^2$ fails to consistently improve on all LLMs, especially the highly capable models (e.g., DeepSeek-V3). Here, we respectfully argue that **MAD-M$^2$ demonstrates significant scalability, particularly on tasks that match the high capabilities of strong models.**
>
> - **[Significant Improvements on Hard Tasks.]** In Table 1, we can observe that the performance of DeepSeek-V3 on AIME24, a highly challenging math competition benchmark, is improved from 0.37 to 0.5. **The substantial improvement on such a hard reasoning task, where the problem is complex and prone to hallucinated reasoning chains, demonstrates the effectiveness of MAD-M$^2$**.
> - **[Ceiling Effects on Easy Tasks.]** Meanwhile, in Table 1, we note that **DeepSeek-V3 has already achieved a saturation point (CoT-SC) with the MAD framework**. Thus, improving the performance of such a powerful LLM on easy benchmarks is inherently difficult due to the ceiling effect. Nevertheless, MAD-M$^2$ still pushes the performance boundary to 0.92, demonstrating the effectiveness of our proposed method.
> - **[Explanations from Theoretical Perspective.]** According to our theoretical analyses in Section 2.2, MAD actually performs inconsistency under different problem settings. Specifically, while the base model struggles to infer the correct answers on extremely hard problems (e.g., AIME 25) due to its intrinsic capability limit, Fig. 5 in Appendix E demonstrates that the performance can be improved when the number of agents increases. Such a phenomenon can further demonstrate the scalability of our proposed method.
>
> For your concern about more results on more LLMs and benchmarks, we will provide more results in the updated version of our paper.
>
> > W2. The proposed masking step roughly doubles token consumption in some cases, but the paper provides limited quantitative analysis of efficiency–performance trade-offs.
>
> **Answer:** Thank you for your comments. According to your comment, we summarize your main concern here is: (1) token consumption of our proposed MAD-M$^2$ method; (2) the efficiency-performance trade-off of our proposed method.
>
> **_[Token Consumption.]_** **The main reason for the double token comsuption is that we add `try...except..` modules in our codes, which increase the consumption of tokens due to the several trials**. To count the token consumption of our proposed MAD-M$^2$ method in a fair way, we remove the `try...except...` modules, and reevaluate baselines and our proposed MAD-M$^2$ in our experiments on Qwen-2.5-7B-Instruct. According to our empirical results (see the second table above), the token consumption of MAD-M$^2$ is more than MAD. Specifically, **MAD-M$^2$ consumes 63.5%, 57.7%, 58.6%, and 56.5% more tokens than MAD on MATH, MMLU_Pro, AIME24, and AIME25, respectively. Such a phenomenon is consistent with our theoretical analysis in our paper (see token consumption analysis in Section 3.2)**. According to our results, we find that such a consumption results from the long responses of the LLMs. We will provide detailed discussion about this phenomenon in our updated version.
>
> **_[Efficiency-Performance Analysis.]_** To address your concern regarding the efficiency-performance trade-offs of our proposed MAD-$^2$, we measure the average time of pipelines, including CoT, CoT-SC, MAD, and MAD-M$^2$, processing each samples. The results are reported in as follows.
>
> - **Average Time Consumption Per Sample**
> | **Methods** | **MATH** | **MMLU_Pro** | **AIME24** | **AIME25** |
> | ------ | ------ | ------ | ------ | ------ |
> | **CoT** | 0.32 | 0.30 | 1.04 | 0.92 |
> | **CoT-SC** | 8.26 | 7.09 | 17.24 | 13.13 |
> | **MAD** | 2.14 | 2.74 | 7.37 | 4.90 |
> | **MAD-M$^2$** | 4.02 | 3.91 | 11.88 | 7.12  |
>
> According to the results, we can observe that all test-time scaling reasoning frameworks, including Cot-SC, MAD, and MAD-M$^2$ consumes more time than the simple CoT. Meanwhile, compared to MAD, MAD-M$^2$ consumes about 1.9, 1.4, 1.6, and 1.5 times more time than MAD on each sample. All these quantitative results indicate that (1) test-time scaling reasoning frameworks naturally consume more time due to the multiple sampling process; (2) MAD-M$^2$ consumes more time than MAD due to the adopted "evaluation and masking" phase.

---

> ### Author Response · Authors · 2025-11-23
>
> > W3. The masking mechanism is described at a high level but not deeply analyzed. For instance, how different scoring heuristics or thresholds affect the results remains unclear. The authors are suggested to add more fine-grained ablations (e.g., varying mask strictness).
> >> Q1. How sensitive is MAD-M2 to the rule for generating the binary mask?
>
> **Response:** Thank you for your comments. According to your comments, we summarize your concern here is **the lack of the investigation of the strictness/sensitivity of the memory masking strategy**.
>
> In our proposed MAD-M$^2$ method, we plug an "evaluation and masking" phase between two round of debate rounds. Specifically, at the beginning of each round (except for the initial round), all responses from the previous debate round will be fed into each agents to determine which responses are adopted as context in current debate round. The agents are allowed to output "YES", "NO", and "NOT SURE" towards each responses. Since "NOT SURE" is an ambiguous answer, treating "NOT SURE" answers in different ways is a feasible way to perform ablations on strictness/sensitivity. **Specifically, if we treat "NOT SURE" as "YES", it is equivalent to applying a loose "threshold" on the answers. In contrast, if we treat "NOT SURE" as "NO", it is equivalent to applying a strict "threshold" on the answers.**  Moreover, **we also adopt perplexity as the measure to perform memory masking**. Specifically, with all memories in the previous round, we will mask those memories with higher perplexity since high perplexity usually implies "uncertain answers." The results are reported as follows.
>
> | **Methods** | **MATH** | **MMLU_Pro** | **AIME24** | **AIME25** |
> | ------ | ------ | ------ | ------ | ------ |
> | **MAD** | 0.485$\pm$0.03 | 0.395$\pm$0.03 | 0.1 | 0.033 |
> | **MAD-M$^2$ (Naive/Loose)** | *0.510$\pm$0.01* | *0.410$\pm$0.03* | **0.13** | **0.067**  |
> | **MAD-M$^2$ (Naive/Strict)** | *0.510$\pm$0.01* | *0.410$\pm$0.03* | **0.13** | **0.067**  |
> | **MAD-M$^2$ (PPL)** | 0.488$\pm$0.03 | 0.395$\pm$0.03 | 0.1 | 0.033 |
>
> According to the table, we can observe that **(1) MAD-M$^2$ is not sensitive to the strictness of masking under the "naive" memory masking settings**. After we checked the log, we found that Qwen2.5-7B-Instruct hardly outputs "NOT SURE" answers (about 3 in 100 tests). Thus, we think that such "robustness" is mainly derived from the decisiveness of the LLM. **(2) MAD-M$^2$ that masks memories via perplexity achieves worse performance than the naive MAD-M$^2$.** We think the main reason here is that the perplexity-based masking strategy implicitly assumes that the perplexity is linearly relevant to the correctness, which is not reasonable in some cases.
>
> > W4. Although the paper compares conceptually to S-MAD and S2-MAD, it doesn’t empirically benchmark against them.
>
> **Response:** Thank you for your comments. We think that your suggestions here are reasonable. We agree that S-MAD and S$^2$-MAD should be considered as baselines in our work. Due to the time and computational resource limit, the experiments are still in the queue. We will add these experiments in the later version of our paper.

---

> ### Author Response · Authors · 2025-11-23
>
> > Q2. Did the authors explore partial masking or soft weighting? Could such approaches improve stability?
>
> **Response:** Thank you for your interesting question. In fact, we thought about soft-weighting. However, we finally chose the hard masking strategy. Two aspects are considered here.
>
> **_[Theoretical Consistency.]_** In Section 2.2, we have built a mathematical model for both CoT-SC and MAD frameworks. In our mathematical model, **we demonstrated that the performance of MAD is negatively related to the number of erroneous memories**. Thus, the hard masking strategy well aligns with our theoretical results. If the soft-weighting strategy is applied, the discrete distribution of $N_{\rm e}$ will become a continuous weighted distribution, which will make the theoretical results more complicated.
>
> **_[Robustness against Distraction.]_** For your concern, previous work [Shi et al. 2023] has demonstrated that LLMs are vulnerable to irrelevant or erroneous context. Although soft masking strategies can somehow remove erroneous content, the distraction may remains (e.g., poorly partial masking). However, hard masking can guarantee that the undesirable context is removed absolutely.
>
> - Shi et al. Large language models can be easily distracted by irrelevant context. In *ICML*, 2023.
>
> > Q3. Can the authors provide more comprehensive experiments on more LLM backbones and compare more baseline MAD approaches?
>
> **Response:** Thank you for your questions. According to your question, we will evaluate more MAD baselines on more LLMs. **Due to the limits of time and computational resources, in this rebuttal, we currently report results on Qwen2.5-7B-Instruct with different memory masking strategies.** More results will be available in the updated version of our paper.
>
> According to the first table above, we can observe that MAD-M$^2$ (Naive) outperforms MAD in all cases. Specifically, **MAD-M$^2$ (Naive) achieves 2.5%, 1.5%, 3%, and 3.4% improvements on MATH, MMLU_Pro, AIME24, and AIME25, respectively**. Moreover, **on AIME24 and AIME25, MAD-M$^2$ achieves comparable or even better performance than CoT-SC**. All these experiments demonstrate the effectiveness of our proposed method. Meanwhile, we also notice that MAD-M$^2$ (PPL) achieves comparable results to MAD, which indicates that perplexity is not an appropriate tool for memory masking since it implicitly assumes that the correctness and the perplexity are linearly relevant.

---

### Official Review · Reviewer_u3Fc · 2025-11-02

**Soundness:** 3
**Presentation:** 3
**Contribution:** 2
**Rating:** 4
**Confidence:** 4

**Summary:**

This paper addresses the robustness of multi-agent debate frameworks to wrong memories from previous debate rounds. The authors provide a theoretical analysis showing that MAD performance is closely related to the number of wrong memories, then propose MAD-M², which allows agents to evaluate and mask undesirable memories before reasoning in the next debate round. Experiments on MATH, MMLU-Pro, AIME24/25 show MAD-M² improves over standard MAD, particularly on easy tasks with weaker models (Qwen2.5-72B, Mixtral-8x7B), though gains on hard tasks and stronger models (DeepSeek-V3) are limited

**Strengths:**

The paper identifies a real issue in MAD: agents can be misled by wrong memories from previous rounds, as illustrated in Figure 1 where Agent 1 initially answers correctly but is misled in Round 2 after seeing Agent 2's incorrect response. This is intuitive and well-demonstrated.

MAD-M² is straightforward to implement: agents evaluate memories, generate binary masks, and reason with filtered memories. The method doesn't require additional models or complex infrastructure, making it practical for real deployment.


The theoretical analysis in Section 2.2 provides useful insights. Propositions 2.2 and 2.3 formalize the performance bounds for CoT-SC and MAD, showing how wrong memories (Ne) exponentially degrade performance through the term e^(-αNe). The distinction between easy problem reasoning (EPR) and hard problem reasoning (HPR) settings offers guidance for when memory masking helps.

**Weaknesses:**

MAD-M² shows improvement on MATH (0.90→0.92, +2%) and AIME24 (0.37→0.50), but degrades or maintains performance on MMLU-Pro (0.83→0.83) and AIME25 (0.40→0.40). Meanwhile, MAD-M² shows improvement on MATH (0.90→0.92, +2%) and AIME24 (0.37→0.50), but degrades or maintains performance on MMLU-Pro (0.83→0.83) and AIME25 (0.40→0.40)


Token consumption analysis in Table 1 shows MAD-M² consumes 50-100% more tokens than standard MAD. The cost-benefit ratio is poor, especially when standard MAD already improves factuality and reasoning significantly over single-agent methods

For assumption 2.1, the claim that the probability of correct answers given memories follows e^(-αNe) is asserted without justification. Why exponential decay? Why not linear, polynomial, or other forms?

Hoeffding requires independent samples, but debate rounds are sequential and dependent - each round conditions on previous memories. The analysis doesn't account for this dependency.


Figures 3-4 show that scaling agents help on easy tasks only (consistent with theory), but MAD-M² doesn't consistently outperform MAD even there

**Questions:**

How often do agents correctly identify wrong memories? What's the precision/recall?


Figure 2 shows one cherry-picked example. Could you also provide the failure cases?

---

> ### Author Response · Authors · 2025-11-23
>
> - **Performance on Qwen-2.5-7B-Instruct**
> In this experiment, we evaluate Qwen-2.5-7B on Math, MMLU_Pro, AIME24, and AIME25 datasets. Each case is the average of 4 runs with different seeds. Among all methods, **MAD** denotes the naive multi-agent debate method, **MAD-M$^2$ (Naive)** denotes the multi-agent debate with **subjectively pruning strategy**, and **MAD-M$^2$ (PPL)** denotes the multi-agent debate with **objectively pruning strategy (perplexity)**.
> | **Methods** | **MATH** | **MMLU_Pro** | **AIME24** | **AIME25** |
> | ------ | ------ | ------ | ------ | ------ |
> | **CoT** | 0.470$\pm$0.05 | 0.403$\pm$0.07 | 0.067 | 0.033 |
> | **CoT-SC** | **0.548$\pm$0.03** | **0.420$\pm$0.04** | 0.1 | **0.067** |
> | **MAD** | 0.485$\pm$0.03 | 0.395$\pm$0.03 | 0.1 | 0.033 |
> | **MAD-M$^2$ (Naive)** | *0.510$\pm$0.01* | *0.410$\pm$0.03* | **0.13** | **0.067**  |
> | **MAD-M$^2$ (PPL)** | 0.488$\pm$0.03 | 0.395$\pm$0.03 | 0.1 | 0.033 |
>
> - **Token Consumption**
> | **Methods** | **MATH** | **MMLU_Pro** | **AIME24** | **AIME25** |
> | ------ | ------ | ------ | ------ | ------ |
> | **CoT** | 57613 | 59034 | 30387 | 29907|
> | **CoT-SC** | 352900 | 360260 | 181639 | 173850 |
> | **MAD** | 347143 | 360029 | 179748 | 178399 |
> | **MAD-M$^2$ (Naive)** | 567655 | 567589 | 285043 | 279145  |
> | **MAD-M$^2$ (PPL)** | 347143 | 360029 | 179748 | 178399 |

---

> ### Author Response · Authors · 2025-11-23
>
> Dear Reviewer u3Fc,
>
> Thank you for your time and efforts in reviewing our paper. We really appreciate your constructive and insightful comments and suggestions. According to your concerns and questions, we provide responses as follows.
>
> > W1. MAD-M² shows improvement on MATH (0.90→0.92, +2%) and AIME24 (0.37→0.50), but degrades or maintains performance on MMLU-Pro (0.83→0.83) and AIME25 (0.40→0.40). Meanwhile, MAD-M² shows improvement on MATH (0.90→0.92, +2%) and AIME24 (0.37→0.50), but degrades or maintains performance on MMLU-Pro (0.83→0.83) and AIME25 (0.40→0.40
>
> **Answer:** Thank you for your comments. According to your comment, we summarize that your main concern here is the performance of our proposed MAD-M$^2$.
>
> According to the results reported in our paper, **the performance is improved on *easy* reasoning problem (e.g., MATH and MMLU_Pro) in almost all cases, while achieving comparable performance on *hard* reasoning problem (e.g., AIME)**. **This is consistent with our theoretical  results in Section 2.2.** For your concern regarding the performance on DeepSeek-V3, in fact, MAD-M$^2$ achieves impressive performance on AIME24 (+13%) and MATH (+2%) with evident gaps, which demonstrates the effectiveness of our proposed method. For the failure case, such as AIME25, we think the main reason is that the benchmark is too hard for the LLM to handle it. Thus, the performance maintains here.
>
> To further solve your concern, we conduct more strict evaluations on all datasets adopted in our paper with a new model, Qwen-2.5-7B-Instruct. To make sure the results are solid, all reported results are the average of 4 trials with different seeds. The performance and token consumptions are reported in the tables above.
>
> According to the results reported in the table, we can observe that MAD-M$^2$ (Naive) outperforms MAD in all cases. Specifically, **MAD-M$^2$ (Naive) achieves 2.5%, 1.5%, 3%, and 3.4% improvements on MATH, MMLU_Pro, AIME24, and AIME25, respectively**. Moreover, **on AIME24 and AIME25, MAD-M$^2$ achieves the comparable or even better performance than CoT-SC**. All these experiments demonstrate the effectiveness of our proposed method. Meanwhile, we also notice that MAD-M$^2$ (PPL) achieves comparable results to MAD, which indicates that perplexity is not an appropriate tools for memory masking since it implicitly assumes that the correctness and the perplexity are linear relevant.
>
> > W2. Token consumption analysis in Table 1 shows MAD-M² consumes 50-100% more tokens than standard MAD. The cost-benefit ratio is poor, especially when standard MAD already improves factuality and reasoning significantly over single-agent methods
>
> **Answer:** Thank you for your comments. According to your comment, we notice that your main concern here is the token consumption of our proposed method. **The main reason here is that we add `try...except..` sections in our codes**.
>
> To further figure out the token consumption in our proposed MAD-M$^2$ method, we remove the `try...except...` parts in our experiments on Qwen-2.5-7B-Instruct. According to our empirical results (see the second table above), the token consumption of MAD-M$^2$ is more than MAD. Specifically, **MAD-M$^2$ consumes 63.5%, 57.7%, 58.6%, and 56.5% more tokens than MAD. Such a phenomenon is consistent with our theoretical analysis in our paper (see token consumption analysis in Section 3.2)**. According to our results, we find that such a consumption results from the long responses of the LLMs. We will provide detailed discussion about this phenomenon in our updated version.
>
> For the cost-benefit ratio and the statement "standard MAD already improves factuality and reasoning significantly over single-agent method" mentioned in your concern, could you provide more explanations about them so that we can provide more discussions to help address your concern?

---

> ### Author Response · Authors · 2025-11-23
>
> > W3. For assumption 2.1, the claim that the probability of correct answers given memories follows e^(-αNe) is asserted without justification. Why exponential decay? Why not linear, polynomial, or other forms?
>
> **Answer:** Thank you for your comments. According to your comment, we notice that your main concern here is **the assumption of the probability that LLMs can correct answer the questions with the responses in the previous round (i.e., $e^{-\alpha N_{\rm e}}$)**. In fact, we adopted this form based on three key justifications:
>
> **_[Boundary Conditions.]_** There are two logical boundary conditions when adopting the exponential form probability. On the one hand, in the case that all responses are correct (i.e., the number of erroneous memories is $N_{\rm e}=0$), the probability $e^{-\alpha N_{\rm e}}=1.0$. **This models the ideal scenario where consistent correct context leads to a correct output.** On the other hand, in the case that all responses are incorrect (i.e., the number of incorrect answers is $N_{\rm a}$), the probability approaches $0$ but remains strictly positive (i.e., $e^{-\alpha N_{\rm e}}>0$). **This aligns with the intuition that LLMs always retain a non-zero probability of self-correction, even in noisy context.**
>
> **_[Modeling Multiplicative Interference.]_** We view each incorrect memory as an independent source of interference. In Information Theory, independent failure factors typically degrade performance multiplicatively rather than additively. For example, if we assume that one erroneous memory reduces performance by a factor $\gamma$, then $N_{\rm e}$ erroneous memories would reduce the performance by $\gamma^{N_{\rm e}}$, which is equivalent to $e^{-\alpha N_{\rm e}}$ with $\alpha=-\ln \gamma$.
>
> **_[Mathematical Validity.]_** For the two forms (e.g., linear and polynomial) metioned in your concern, on the one hand, linear forms (e.g., $1-\beta N_{\rm e}$) will inevitably result in **negative probability** when $N_{\rm e}$ is large, which violates the axioms of probability. On the other hand, polynomial forms often require complex constraints with $[0, 1]$ interval and lack the intuitive property of constant proportional decay rate.
>
> > W4. Hoeffding requires independent samples, but debate rounds are sequential and dependent - each round conditions on previous memories. The analysis doesn't account for this dependency.
>
> **Answer**: Thanks for your comments. According to your comment, we notice your concern mainly focuses on whether the Hoeffding Inequality can be applied in our proof. Note that we **did not assume independence across rounds**. **The only independence we assume is within a single round, conditional on the history**.
>
> We agree that debate rounds are sequential and therefore **responses are not i.i.d. _across rounds_**. However, in our theoretical analysis, the **Hoeffding Inequality is only applied within a fixed round to aggregate independent response samplings**. Specifically, in Round 1, given the prompt, $N_{\rm a}$ responses are independently generated from different agents/decoding randomness. In Round 2, $N_{\rm a}$ responses are also **independently** generated from agents conditioned on the response history.

---

> ### Author Response · Authors · 2025-11-23
>
> > W5. Figures 3-4 show that scaling agents help on easy tasks only (consistent with theory), but MAD-M² doesn't consistently outperform MAD even there.
>
> **Answer:** Thank you for your comments. According to your concern, **we would like to clarify that the comparable performance between MAD-M$^2$ and MAD on easy tasks is actually a normal behavior**.
>
> The main idea of MAD-M$^2$ is filtering out erroneous memories to reduce the noise in the memories. However, two cases may happen. On the one hand, in the easy reasoning problem, the noisy memories that can be easily detected and modified may be few since a consensus can be formed in most cases. On the other hand, some erronesou memories (e.g., wrong but convincing logics) may be too hard to be detected. Thus, MAD-M$^2$ sometimes achieves comparable performance to MAD. Meanwhile, introducing more agents may also further result in more noisy. Thus, the performance may flunctuate.
>
> > Q1. How often do agents correctly identify wrong memories? What's the precision/recall?
>
> **Answer:** Thank you for your question. To solve your concern, we conduct experiments with Qwen2.5-7B-Instruct on all benchmarks adopted in our paper. Specifically, we investigate the memories and the masks of the memories in the first round. Here, we provide two standards to measure the capability of agents in identifying the erroneous memories.
> - **Strict case:** All erroneous memories in the debate round are detected.
> - **Loose case:** At least one erronesou memory in the debate round is detected.
>
> In both two cases, if the correct memories are removed, the debate round will be considered as the failure case. The results are reported as follows.
>
> | **Standard** | **MATH** | **MMLU_Pro** | **AIME24** | **AIME25** |
> | ------ | ------ | ------ | ------ | ------ |
> | **Strict** | 32.0% | 7.4% | 90.0% | 96.67%|
> | **Loose** | 43.8% | 33.4% | 90.0% | 96.67% |
>
> According to the results in the table, we can obtain the following observations.
>
> **_[Task Sensitivity.]_** According to the results above, we find that Qwen2.5-7B-Instruct performs better in erroneous memory detection in the math reasoning problems than that in the language understanding reasoning problems.
>
> **_[Abnormal Performance in AIME.]_** According to the results, we find that Qwen2.5-7B-Instruct performs better in erroneous memory detection on hard math reasoning problems (AIME) than easy ones (MATH). Specifically, almost all erroneous memories in AIME tasks are detected while the performance on this benchmark is poor.
>
> The reason for such a phenomenon may be that LLMs usually generate reasoning paths with poor logical structures on such hard benchmarks due to the complexity of the problem. Compared to response generation, identifying the errors is much easier. Thus, LLMs can easily detect the erroneous reasoning with poor logical structures but fails to generate correct ones.
>
> > Q2. Figure 2 shows one cherry-picked example. Could you also provide the failure cases?
>
> **Answer:** Thanks for your comments, we agree that it is necessary to provide both success and failure cases in the paper to evaluate our proposed method in a comprehensive way. We will update these cases in the updated version of our paper.

---

### Meta-Review · Area_Chair_k3cz · 2026-01-07

**Summary:**

This paper proposes to augment multiagent debate with a memory masking system, where incorrect answers are identified and masked from the debate context. The paper mathematically shows how this can improve the performance of a multi-agent debate system and empirically also illustrates this across a set of models and benchmarks. Reviewers had some concerns about the marginal performance gains, theoretical results, and additional analysis that the authors provided. As a result, I believe the paper is boarderline and can be accepted to the conference.

**Reviewer Concerns:**

Please see the next section for detailed reviewer concerns. Overall, reviewers were concerned about the marginal performance gains, the underlying mathematical assumption, baseline comparisons, and wanted some additional analysis of the method. The authors provided a substantial rebuttal with significant change in the text of the paper.

**Reviewer Scores:**

I believe that the authors have mostly addressed the major concerns of each reviewer.

Reviewer u3Fc had concerns about the marginal gains of the proposed method on several tasks. The authors clarified that this is normal for simple problems where the majority of answers are correct. In addition, the reviewer asked about the analysis of the accuracy of error detection which the authors provided. In addition, the authors clarified their mathematical assumptions. As a result, I believe the reviewer would have increased their score to a marginal accept.

Reviewer  7GkP had concerns about the marginal gains of the approach on more capable models, want more analysis on the masking pattern, and asked for additional baseline comparisons.  The reviewers clarified the gains of DeepSeek on more complex datasets, provided additional analysis of masking, and promised to add the additional MAD baseline method comparisons (which the AC urges the authors to do). As a result, I believe this reviewer would have also increased their score to a marginal accept.

Reviewer gCX4 had concerns about the theoretical nature of the paper as well as the limited gains over self-consistency. The authors were able to successfully explain the nature of the empirical results in the paper. In addition, the authors clarified that the method outperformed MAD and in harder datasets actually also outperformed self-consistency. Given this, the AC believes reviewer gCX4 would have also improved their score to a marginal accept. At the same time, the AC encourages the reviewers to evaluate their approach on some additional harder datasets to illustrate how MAD-M2 can actually more consistently outperform self-consistency. In addition, observation 2 in Main Results is a bit contradictory to the message of the paper, and the AC suggests removing or qualifying the statement, as if self-consistency is always better than the proposed approach, it makes limited sense to have the paper.

---

### Decision · Program_Chairs · 2026-01-26

Accept (Poster)